**EMBO** *reports*

# LATS1/2 inactivation in the mammary epithelium drives the evolution of a tumor-associated niche

Joseph G Kern[1], Lina Kroehling [2,3], Anthony J Spinella [1], Stefano Monti[2,3,4] & Xaralabos Varelas [1✉]

## Abstract

**Basal-like breast cancers exhibit distinct cellular heterogeneity that contributes to disease pathology. In this study we used a genetic mouse model of basal-like breast cancer driven by epithelial-specific inactivation of the Hippo pathway-regulating LATS1 and LATS2 kinases to elucidate epithelial-stromal interactions. We demonstrate that basal-like carcinoma initiation in this model is accompanied by the accumulation of distinct cancer-associated fibroblasts and macrophages and dramatic extracellular matrix remodeling, phenocopying the stromal diversity observed in human triple-negative breast tumors. Dysregulated epithelial-stromal signals were observed, including those mediated by TGF-β, PDGF, and CSF. Autonomous activation of the transcriptional effector TAZ was observed in LATS1/2-deleted cells along with non-autonomous activation within the evolving tumor niche. We further show that inhibition of the YAP/TAZ-associated TEAD family of transcription factors blocks the development of the carcinomas and associated microenvironment. These observations demonstrate that carcinomas resulting from Hippo pathway dysregulation in the mammary epithelium are sufficient to drive cellular events that promote a basal-like tumor-associated niche and suggest that targeting dysregulated YAP/TAZ-TEAD activity may offer a therapeutic opportunity for basal-like mammary tumors.**

**Keywords** Tumor Microenvironment; LATS1/2; YAP/TAZ; Hippo Signaling; TEAD
**Subject Categories** Cancer; Immunology; Signal Transduction

## Introduction

Tumors are composed of intricate cell ecosystems that cooperate to promote oncogenic growth and progression (Valkenburg et al, 2018). The importance of the tumor microenvironment is becoming increasingly clear and has been the focus of a growing number of treatment avenues (Valkenburg et al, 2018). Endothelial changes that facilitate angiogenesis have long been appreciated

(Ruoslahti, 2002), and alterations in stromal mesenchymal cells such as the emergence of cancer-associated fibroblasts (CAFs) promotes tumorigenesis through reciprocal signaling with cancer cells and other stromal cells (Kalluri, 2016). The tumor niche also influences oncogenic behavior through the mediation of immune evasion (Gonzalez et al, 2018). A myriad of distinct stromal phenotypes has been identified in the mammary gland and in breast cancer, highlighting the complexity of such stromal changes and the need to better understand their etiology (Eiro et al, 2019; Inman et al, 2015; Mao et al, 2013).

Breast tumors are composed of complex stromal populations that evolve distinctly in individual tumors (Tu and Karnoub, 2022). Triple-negative breast cancer (TNBC) is a breast cancer subtype that exhibits notable remodeling of stromal cells and neighboring extracellular matrix (ECM) (Acerbi et al, 2015; Costa et al, 2018; Wu et al, 2021; Wu et al, 2020; Zagami and Carey, 2022), and most, but not all, TNBC tumors exhibit basal-like molecular traits (Perou, 2011). Stromal accumulation in TNBC tumors informs treatment responses and predicts poor prognosis (van der Spek et al, 2020). Molecular signals that mediate and respond to stromal remodeling are poorly understood, and thus, a better understanding of the interactions between tumor cells and the stromal microenvironment may offer the opportunity to uncover druggable targets that mediate both recruitment and oncogenic phenotypes of stromal cells.

The Hippo signaling pathway is heavily implicated in promoting cellular plasticity and oncogenic properties in breast tumors (Kern et al, 2022; Panciera et al, 2016; Zanconato et al, 2016). We and others have demonstrated important roles for the Hippo pathway transcriptional effectors YAP and TAZ (YAP/TAZ) in promoting basal-like and triple-negative breast cancer development (Kern et al, 2022; Kim et al, 2015; Soyama et al, 2022). Dysregulation of the Hippo pathway kinases LATS1 and LATS2 (LATS1/2) drives aberrant nuclear YAP/TAZ activity, which in the mammary epithelium promotes luminal-basal plasticity and rapid cellular overgrowth (Kern et al, 2022). YAP/TAZ localization can be influenced by tissue stiffness (Dupont et al, 2011), suggesting an association between YAP/TAZ activity and stiffness related changes in a tumor microenvironment. However, despite the influence of Hippo pathway signaling on tumor development, how dysregulated Hippo pathway signaling in the mammary epithelium shapes an evolving tumor microenvironment is unknown.

In this study we investigated how aberrant activation of YAP/TAZ in the mammary epithelium impacts the stromal

[1]Department of Biochemistry and Cell Biology, Boston University Chobanian & Avedisian School of Medicine, Boston, MA 02118, USA. [2]Department of Medicine, Computational Biomedicine Section, Boston University Chobanian & Avedisian School of Medicine, Boston, MA 02118, USA. [3]Bioinformatics Program, Boston University, Boston, MA 02215, USA. [4]Department of Biostatistics, Boston University School of Public Health, Boston, MA 02118, USA. ✉E-mail: xvarelas@bu.edu

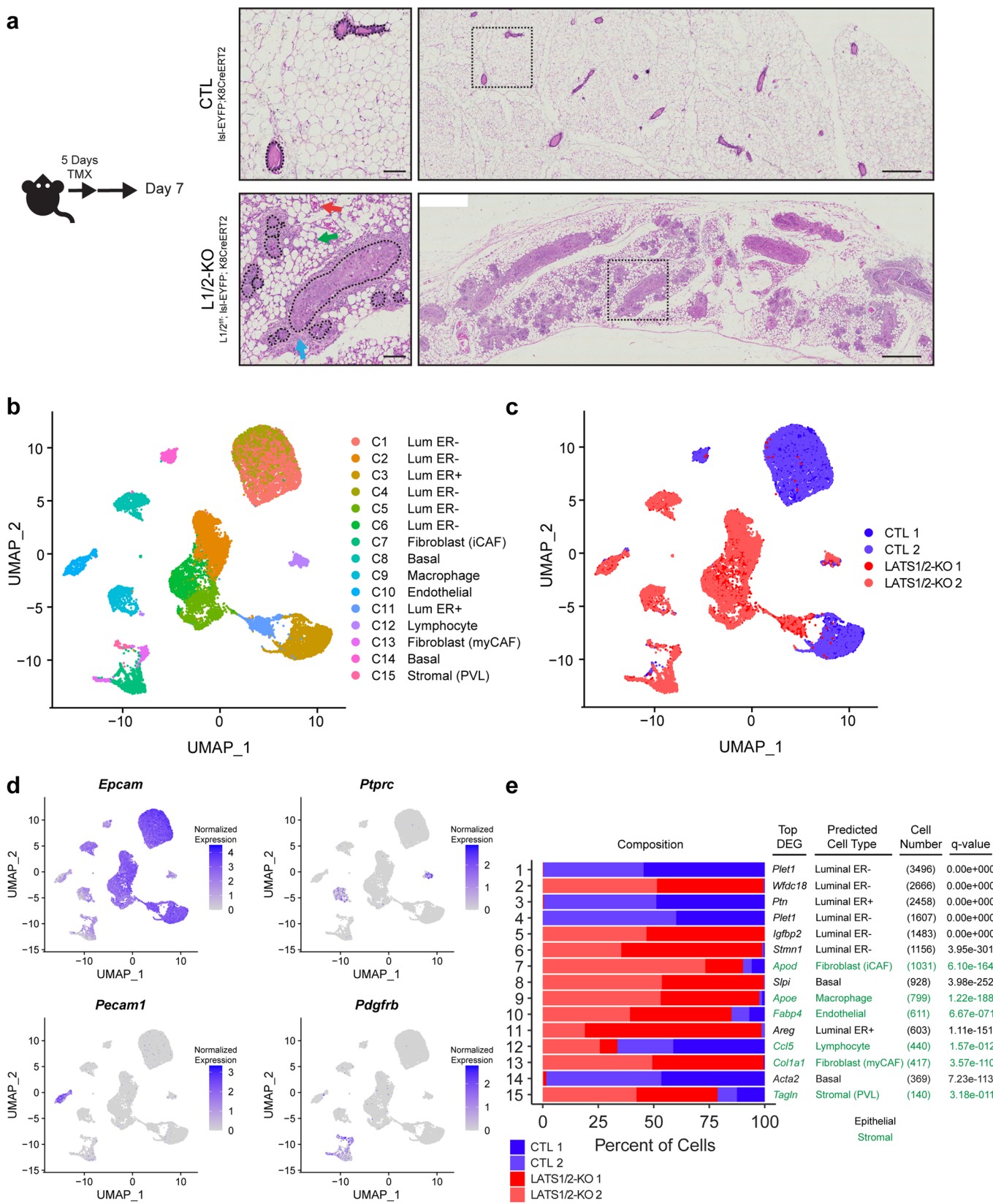

Figure 1.   Epithelial and stromal alterations in basal-like mammary carcinomas driven by LATS1/2 deletion.

(A) Hematoxylin and Eosin stain of CTL and LATS1/2-KO mammary glands. Shown on the left are subset images of the regions outlined by dashed boxes on the right. Outlines of ducts were drawn using black dotted lines in the left images by approximating the location of the basement membranes. Blue arrow indicates an example of stromal cells surrounding ducts. Red arrow indicates an example of a blood vessel. Green arrow indicates an example of cells accumulating between adipocytes. (Scale bar, 100 μm left images; 500 μm right images) ($n = 9$ CTL mice, $n = 9$ L1/2-KO mice). (B) UMAP of clusters identified from single-cell RNA-sequencing of CTL (LATS1/2$^{ff}$; noCre) ($n = 2$) and LATS1/2-KO (LATS1/2$^{ff}$; lsl-EYFP; Krt8CreERT2) ($n = 2$) mammary glands. (C) UMAP of single-cell RNA-sequencing clusters colored by condition. (D) UMAPs of transcriptional markers of epithelial, fibroblast, endothelial, and immune cell populations. (E) Composition of single-cell RNA-sequencing clusters colored by condition, along with the top differentially expressed gene in every cluster relative to all other clusters (ranked by Log2FC and using genes expressed in at least 50% of cells in the cluster), predicted cell type, number of cells in each cluster, and q-values derived from Fisher's Test demonstrating differences in cluster compositions. Cluster annotations are colored according to epithelial and stromal cell identities. Composition percentages are normalized to the total cell counts per sample. Source data are available online for this figure.

microenvironment, examining stromal-epithelial communication in the evolution of the basal-like tumor niche. We show that basal-like mammary carcinomas initiated by deletion of Lats1/2 in luminal mammary epithelial cells is sufficient to promote stromal cell remodeling and ECM accumulation resembling the phenotypes observed in human triple-negative tumors. Through single cell RNA-sequencing and histological analysis we uncover major signaling axes between epithelial and stromal cell populations within Hippo pathway-dysregulated basal-like mammary tumors that result in cell autonomous and non-autonomous oncogenic cues. We demonstrate that targeting dysregulated YAP/TAZ activity using a small molecule inhibitor of the TEAD transcription factors, which are key mediators of YAP/TAZ-regulated gene expression (Chan et al, 2009; Zhao et al, 2008), prevents the development of tumors driven by Lats1/2-deletion in the mammary epithelium, including blocking the onset of the tumor-stromal niche. Our observations indicate that Hippo pathway inactivation drives the evolution of a basal-like mammary tumor microenvironment through discrete cell communication mechanisms, and further suggest that targeting dysregulated YAP/TAZ activity through the use of emerging TEAD family inhibitors may offer a therapeutic opportunity for basal-like mammary tumors.

## Results

### Single cell RNA-sequencing identifies distinct epithelial and stromal changes that accompany basal-like mammary carcinoma development resulting from Lats1/2 inactivation

We previously demonstrated that deletion of the *Lats1* and *Lats2* genes in luminal mammary cells induces luminal-basal plasticity and YAP/TAZ-mediated development of basal-like mammary carcinomas (Kern et al, 2022). For this, we developed a genetic mouse model allowing conditional luminal epithelial cell-specific deletion of *Lats1/2* using a Tamoxifen-inducible Cre recombinase under the control of the *Krt8* promoter, which we combined with a loxP-stop-loxP (lsl)-EYFP lineage trace to mark Cre$^+$ cells (Lats1/2$^{ff}$; lsl-EYFP; Krt8CreERT2). The phenotype driven by Tamoxifen-induced Lats1/2 knockout in this system resembles basal-like ductal carcinoma in situ (DCIS) and is marked by rapid overgrowth of basal-like cells within mammary ducts (Kern et al, 2022). Interestingly, we observed that the progression of this overgrowth phenotype was accompanied by a dramatic remodeling of the local mammary stroma and influx of stromal cells (Figs. 1A and EV1A).

This remodeling included an accumulation of stromal cells adjacent to the mammary ducts of Lats1/2$^{ff}$; lsl-EYFP; Krt8CreERT2 mice, blood vessels near many ducts with a DCIS-like phenotype, and alterations to the fat pad including notable cellularity between adipocytes.

The reorganization of the stromal compartment in Lats1/2$^{ff}$; lsl-EYFP; Krt8CreERT2 mice raised questions about the composition of these cell populations and their potential relationship to carcinoma development. To elucidate the changes in mammary stromal and epithelial cell populations of Lats1/2$^{ff}$; lsl-EYFP; Krt8CreERT2 mice, we isolated cells from the mammary glands of control (Lats1/2$^{ff}$; no Cre) and Lats1/2-KO (Lats1/2$^{ff}$; lsl-EYFP; Krt8CreERT2) mice and performed single-cell RNA-sequencing. Samples were sorted for viability, but no additional measures were used to isolate cell types prior to collectively running the samples on the 10X Genomics platform. We sequenced cells from two mice per condition and analyzed between 3515 to 5274 total cells per mouse after applying quality control as described in the Methods. PC analysis based on the top 3000 highly variable genes identified by SCtransform was performed and cells were clustered based on the Louvain graph-based method applied to the top 20 PCs. Uniform Manifold Approximation and Projection (UMAP) was then used to visualize the resultant cell clusters (Fig. 1B). There were significant differences in cell clustering dependent on condition (Fig. 1C). Using defined gene markers for epithelial (*Epcam*), immune (*Ptprc*), endothelial (*Pecam1*), and fibroblast/perivascular (*Pdgfrb*) cells, we found that each of these populations were represented in our data (Fig. 1D). Further analysis revealed specific marker genes differentially expressed in each cluster, which we used to manually annotate clusters by cell type (Figs. 1E and EV1B). Of note, mature adipocytes were not identified, likely due to the difficulty of isolating and capturing these cells for single-cell sequencing (Van Hauwaert et al, 2021). Many of the identified clusters showed compositional differences between conditions and were nearly entirely represented in only one of the two conditions, highlighting strong differences in cellular phenotypes in the Lats1/2$^{ff}$; lsl-EYFP; Krt8CreERT2 mice relative to controls.

### Lats1/2-null epithelial cells show distinct signaling to stromal cells

As expected, we observed several changes in mammary epithelial cell phenotypes resulting from Lats1/2 inactivation. To better define luminal and basal cell lineages in our data, we used previously identified markers of luminal hormone receptor-positive (HR+), luminal hormone receptor negative (HR-), and basal cells to

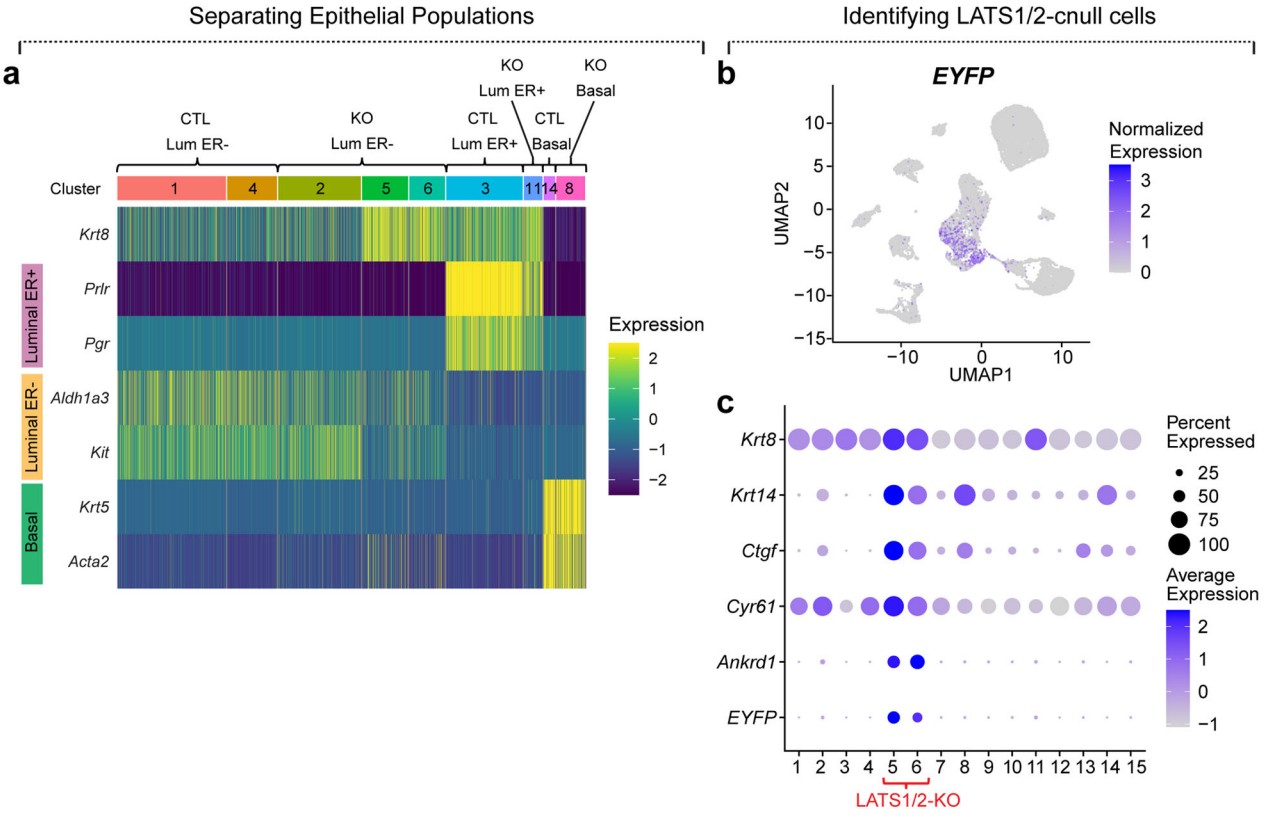

**Figure 2.  Characterization of epithelial-derived signaling pathways.**

(A) Heatmap of marker genes of luminal and basal mammary cell fate across epithelial clusters, along with designation of epithelial cell type, predominant condition represented in each cluster, and revised groupings of epithelial cells. (B) UMAP of *EYFP* expression across all clusters. (C) Dot heatmap of transcriptional markers of luminal-basal plasticity, YAP/TAZ targets, and *EYFP*. (D) CellChat circle plots showing communication networks between epithelial cell groups (senders) and stromal cell groups (receivers) in CTL and LATS1/2-KO mammary glands. Line widths represent the total number of interactions identified. (E) Heatmap of communication probabilities of signaling pathways from epithelial groups to stromal groups in CTL and LATS1/2-KO mammary glands.

differentiate these populations (Fig. 2A) (Bach et al, 2017). We were able to assign clusters from both treatment conditions to each of these three epithelial lineages. The distinct epithelial clusters in this model allowed us to group and rename the epithelial cell types according to their cell lineage and predominant condition: Control (CTL) Luminal Estrogen Receptor negative (ER-), Knockout (KO) Luminal ER-, CTL Luminal ER positive (ER+), KO Luminal ER+, CTL Basal, and KO Basal (Fig. 2A). As the luminal ER- cell population represented multiple clusters for each condition, we grouped these together to give one CTL Luminal ER- and one KO Luminal ER- group.

We next sought to define cells that aligned with Cre activity in our Lats1/2$^{ff}$; lsl-EYFP; Krt8CreERT2 mice. Analysis of *EYFP* expression across all clusters revealed predominant *EYFP* expression present in Clusters 5 and 6, which we had assigned to the KO Luminal ER- group (Figs. 2B and EV1C). This observation is consistent with prior findings that basal-like carcinomas share characteristics of ER- luminal progenitor cells, and that ER- luminal progenitors can serve as an origin of basal-like carcinomas (Kern et al, 2022; Lim et al, 2009; Molyneux et al, 2010). As we reported that Lats1/2-null cells exhibit luminal-basal plasticity and strong YAP/TAZ-driven signatures in the Lats1/2$^{ff}$; lsl-EYFP; Krt8CreERT2 model, we further confirmed the presence of Lats1/2-null cells by analyzing expression of *Krt8*, *Krt14*, and the YAP/TAZ targets *Ctgf*, *Cyr61*, and *Ankrd1* across all cell clusters, observing strong co-expression of these genes in Clusters 5 and 6 (Fig. 2C) (Kern et al, 2022).

As the carcinomas that form in this model are driven by epithelial-specific inactivation of Lats1/2, we hypothesized that the epithelium must signal to other cell populations to elicit the observed stromal changes. To define such signals in our single-cell RNA-sequencing dataset we used CellChat (Jin et al, 2021), a tool that integrates expression of ligands, receptors, and cofactors in single-cell RNA-sequencing data with prior knowledge of signaling patterns to predict crosstalk between cell groups. We defined thirteen cell groups in our dataset which we used for all CellChat analyses in this study (Fig. EV1D), including six epithelial groups according to the groups defined in Fig. 2A, as well as unique groups for each fibroblast/stromal cluster (C7, C13, and C15), macrophages (C9), endothelial cells (C10), and lymphocytes (C12). We additionally designated a separate group for epithelial cells expressing *EYFP* in the Lats1/2 KO samples to mark Cre active Lats1/2-KO cells. We performed all CellChat analyses in this study separately between control and Lats1/2-KO samples to avoid false positive signaling results between the two conditions. The results of all CellChat analyses performed between cell populations can be viewed in Dataset EV1.

We first sought to agnostically determine the major signals between epithelial cells and all stromal cell types in our data (Fig. 2D,E). This analysis identified signaling axes between epithelial groups and every

stromal cell type. Some shared epithelial-stromal communication pathways were observed between CTL and Lats1/2-KO samples, including strong representation of Thrombospondin (THBS) from Lum ER- cells, and Collagen from basal cells. However, these pathways showed stronger communication scores in the Lats1/2-KO samples relative to control, suggesting amplification of these signals in Lats1/2-dysregulated epithelia. Several shared pathways were observed in EYFP+ (Cre active) and EYFP- (Cre inactive) Lats1/2-KO epithelial cells, including Epidermal Growth Factor (EGF), Platelet derived growth factor (PDGF), Junctional Adhesion Molecule (JAM), Periostin, Fibronectin-1 (FN1), Transforming Growth Factor-beta (TGFb) and Calcitonin receptor (CALCR) signaling. However, EYFP+ (Cre-active) Lats1/2-KO epithelial cells also exhibited several unique pathways including Semaphorin-3 (SEMA3), Colony stimulating factor (CSF), and Ephrin-B (EPHB) signaling. The identification of these prompted us to further interrogate specific pathways mediating stromal alterations that are driven by Hippo inactivation.

## Carcinomas driven by Lats1/2 inactivation display increases in cancer-associated fibroblast populations that share characteristics with human tumors

We noted that CellChat analysis revealed interactions between *EYFP*$^+$ (Cre active) cells and multiple fibroblast-like cell populations (Fig. 2D,E), leading us to explore these populations further. Immunofluorescence staining for Platelet Derived Growth Factor Receptor β (PDGFRβ), a marker of fibroblasts and perivascular cells (Kalluri, 2016), demonstrated increases in PDGFRβ$^+$ cells in the stroma of Lats1/2-KO mice, with notable presence surrounding mammary ducts (Figs. 3A and EV2A). In addition, staining for αSMA, a marker of fibroblast-to-myofibroblast activation (Kalluri, 2016), revealed the presence of αSMA$^+$PDGFRβ$^+$ fibroblasts immediately adjacent to the basement membrane in Lats1/2$^{ff}$; lsl-EYFP; Krt8CreERT2 mice, suggesting myofibroblast presence in these mammary glands (Fig. 3B).

We observed several separate clusters of stromal cells with fibroblast-associated markers in our single-cell RNA-sequencing data, specifically clusters 7, 13, and 15. Differential expression analysis on these three clusters revealed specific genes enriched in each population (Fig. 3C). Cluster 7 exhibited a mix of genes, several associated with immune and inflammatory-related processes. Cluster 13 strongly expressed ECM-associated genes. Cluster 15 expressed genes associated with a smooth muscle cell and vascular cell state, suggesting a vascular-associated identity for these cells. Prior studies have identified stromal cell populations with distinct gene expression programs in human breast cancer (Costa et al, 2018; Kieffer et al, 2020; Wu et al, 2021; Wu et al, 2020), so we wondered whether the populations we observed resemble those in human disease. Four distinct populations of tumor-associated CAF and stromal cells have been described in human TNBC: myofibroblast-like CAFs (myCAFs), inflammatory

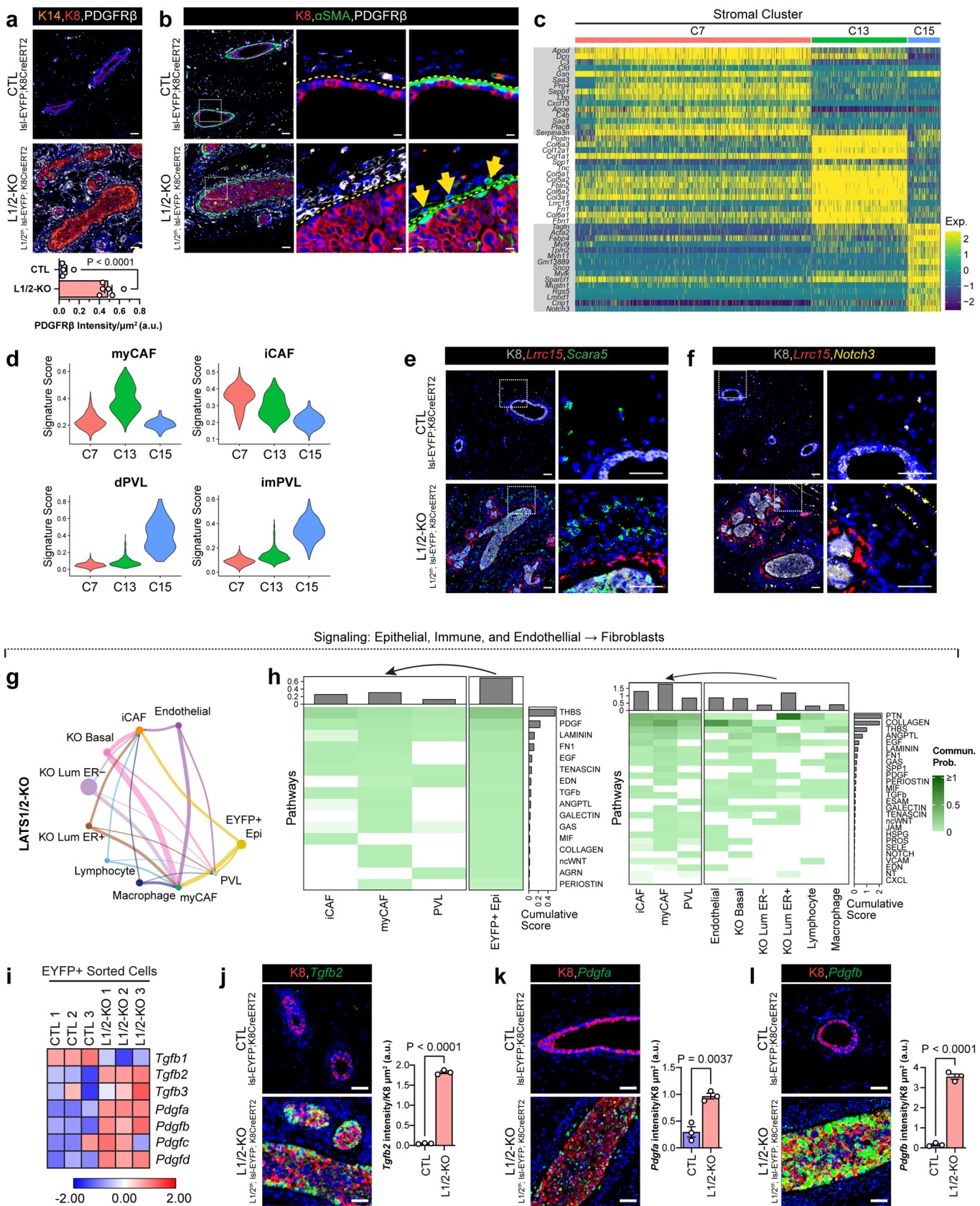

Figure 3. Accumulation of cancer-associated fibroblasts in the mammary stroma of mice with LATS1/2 inactivation.

(A) Immunofluorescence staining for KRT14, KRT8, and PDGFRβ in CTL and LATS1/2-KO mammary glands (Scale bar, 50 μm) and quantification of PDGFRβ intensity/area (n = 6 CTL mice, n = 6 L1/2-KO mice, 5 regions analyzed per mouse. Data are shown with mean ± SEM. Two-tailed unpaired T-test). (B) Immunofluorescence staining for KRT8, αSMA, and PDGFRβ in CTL and LATS1/2-KO mammary glands. Shown on the right are subset images of the regions outlined by the dashed boxes on the left. Yellow dotted lines are drawn to approximate the location of the basement membrane. Yellow arrows indicate αSMA⁺PDGFRβ⁺ cells (Scale bar: 50 μm left images; 10 μm right images) (n = 3 CTL mice, n = 3 L1/2-KO mice). (C) Heatmap of top differentially expressed genes (ranked by Log2FC and expressed in >0 percent of cells) between three fibroblast/stromal clusters represented in single-cell RNA-sequencing data. (D) Correlation of human TNBC CAF and stromal signatures to signatures of stromal populations identified via single-cell RNA-sequencing on CTL and LATS1/2-KO mammary glands (n = 1031 cells for C7, n = 417 cells for C13, n = 140 cells for C15). (E) RNAscope of Lrrc15 and Scara5 along with IF for KRT8 in CTL and LATS1/2-KO mammary glands. Shown on the right are subset images of the regions outlined by the dashed boxes on the left (Scale bar 50 μm) (n = 3 CTL mice, n = 3 L1/2-KO mice). (F) RNAscope of Lrrc15 and Notch3 along with IF for KRT8 in CTL and LATS1/2-KO mammary glands. Shown on the right are zoomed images of the regions outlined by the dashed boxes on the left (Scale bar 50 μm) (n = 3 CTL mice, n = 3 L1/2-KO mice). (G) CellChat circle plots showing communication networks from epithelial and stromal cell groups (senders) to iCAF, myCAF, and PVL groups (receivers) in LATS1/2-KO mammary glands. Line width represents the total number of interactions identified. (H) Heatmap of communication probabilities of signaling pathways from epithelial and stromal cell groups to iCAFs, myCAFs, and PVL cells in LATS1/2-KO mammary glands. (I) RNA-sequencing expression of PDGF and TGF-β ligands in sorted EYFP⁺ cells from CTL (lsl-EYFP; Krt8CreERT2) and LATS1/2-KO (LATS1/2ff; lsl-EYFP; Krt8CreERT2) mouse mammary glands. (J–L) RNAscope for Tgfb2 (J), Pdgfa (K), and Pdgfb (L) in CTL and LATS1/2-KO mammary glands, along with IF for KRT8 (Scale bars 50 μm) and quantification of Tgfb2, Pdgfa, and Pdgfb intensity/K8+ area (n = 3 CTL mice, n = 3 L1/2-KO mice each for (J–L), 5 regions analyzed per mouse. Data are shown with mean ± SEM. Two-tailed unpaired T-tests). Source data are available online for this figure.

CAFs (iCAFs), differentiated perivascular-like cells (dPVLs), and immature perivascular-like cells (imPVLs) (Wu et al, 2020), all named based on their patterns of differentially expressed genes and similarities to pancreatic cancer CAFs described in other studies (Biffi et al, 2019; Elyada et al, 2019; Ohlund et al, 2017). To assess the similarity of these human TNBC stromal populations to those we observe in Lats1/2ff; lsl-EYFP; Krt8CreERT2 mice, we compared the transcriptional signatures of each (Wu et al, 2020) to those of our own cell populations (Fig. 3D), which revealed that each of the fibroblast clusters we observed in the Lats1/2-null microenvironment correlated with a distinct human cell population. Cluster 7 displayed a strong enrichment for iCAF genes, Cluster 13 was enriched for myCAF genes, and Cluster 15 was enriched for PVL genes. We therefore labeled these clusters "iCAF", "myCAF", and "PVL" in our single-cell RNA-sequencing data.

To validate the presence of these separate CAF/stromal subpopulations and view their spatial distribution in the mammary glands of Lats1/2ff; lsl-EYFP; Krt8CreERT2 mice, we identified unique genes highly associated with each population: Scara5 to mark iCAFs, Lrrc15 to mark myCAFs, and Notch3 to mark PVL cells (Fig. EV2B–D). These genes are also conserved in human TNBC iCAF, myCAF, and PVL populations (Kieffer et al, 2020; Wu et al, 2020). We then performed multiplexed RNAscope in situ hybridization on tissues from control and Lats1/2ff; lsl-EYFP; Krt8CreERT2 mice using probes specific for Lrrc15, Scara5, and Notch3 (Figs. 3E,F and EV2E,F). This revealed distinct spatial patterns of these transcripts and their indicated cell types. We observed a striking presence of myCAFs immediately adjacent to the mammary ducts and developing carcinomas in Lats1/2-KO mice, whereas iCAFs and PVLs were dispersed throughout the stroma farther from ducts. iCAFs were interspersed throughout adipocytes, and PVL cells were often found surrounding blood vessels, supporting the PVL identity as perivascular-like. Together, this data demonstrates that CAFs present in the stroma of mammary carcinomas driven by Lats1/2 inactivation share similarities to CAFs in human TNBC, and that stromal CAF alterations develop during early initiation of basal-like carcinomas.

To determine potential mechanisms of CAF recruitment in Lats1/2ff; lsl-EYFP; Krt8CreERT2 mice, we next performed a CellChat analysis identifying signals being received by the iCAF, myCAF, and PVL populations from other cell groups (Figs. 3G,H

and EV2G,H). The largest amount of signaling in LATS1/2-KO mice was predicted to occur towards myCAFs, and we identified several pathways signaling to the CAF populations, many of which were ECM-associated. Notably, the PDGF and TGF-β pathways were predicted as key signaling axes between EYFP⁺ cells and the CAF subclusters. In addition, RNA-sequencing from EYFP-sorted cells isolated from Lats1/2ff; lsl-EYFP; Krt8CreERT2 mice (Kern et al, 2022) indicated noticeable upregulation of ligands from both the PDGF family and TGF-β family in cells with Lats1/2 inactivation relative to control cells (Fig. 3I). The TGF-β and PDGF signaling pathways are established drivers of fibroblast activation and recruitment (Kalluri, 2016). Therefore, the upregulation of these signals in Lats1/2-null tumor cells suggest that they contribute to the recruitment and activation of the CAF populations observed in the stroma of these mice. To confirm expression of PDGF and TGF-β ligands in Lats1/2-null mammary carcinomas, we performed RNAscope in situ hybridization. This demonstrated robust increases in expression of Tgfb2, Pdgfa, and Pdgfb in the mammary glands of Lats1/2-KO mice, predominantly in epithelial carcinoma cells (Figs. 3J–L and EV2I–K).

## Distinct extracellular matrix changes accompany tumorigenesis in Lats1/2-null mammary carcinomas

We next sought to define the major functions of iCAFs, myCAFs, and PVLs in Lats1/2-null carcinomas. To do this, we performed CellChat analysis designating the iCAF, myCAF, and PVL groups as signal-sending groups and all other groups as receivers (Figs. 4A,B and EV3A,B). Of the pathways enriched in this analysis, we noticed a strong representation of ECM-associated pathways signaling from CAFs to other cell types. The myCAF group in particular showed strong communication through ECM pathways such as COLLAGEN, LAMININ, THBS, FN1, and TENASCIN, among others. This is in line with prior studies showing a strong ECM-associated signature in myCAF populations (Wu et al, 2020). Of note, many of these ECM pathways were also present at a lower level in the iCAF and PVL groups.

ECM changes are known to influence tumorigenic properties such as stem cell and invasive phenotypes (Kalluri, 2016). As our analysis revealed strong ECM signaling from CAF populations to other cell types, we explored the ECM phenotypes elicited by CAFs

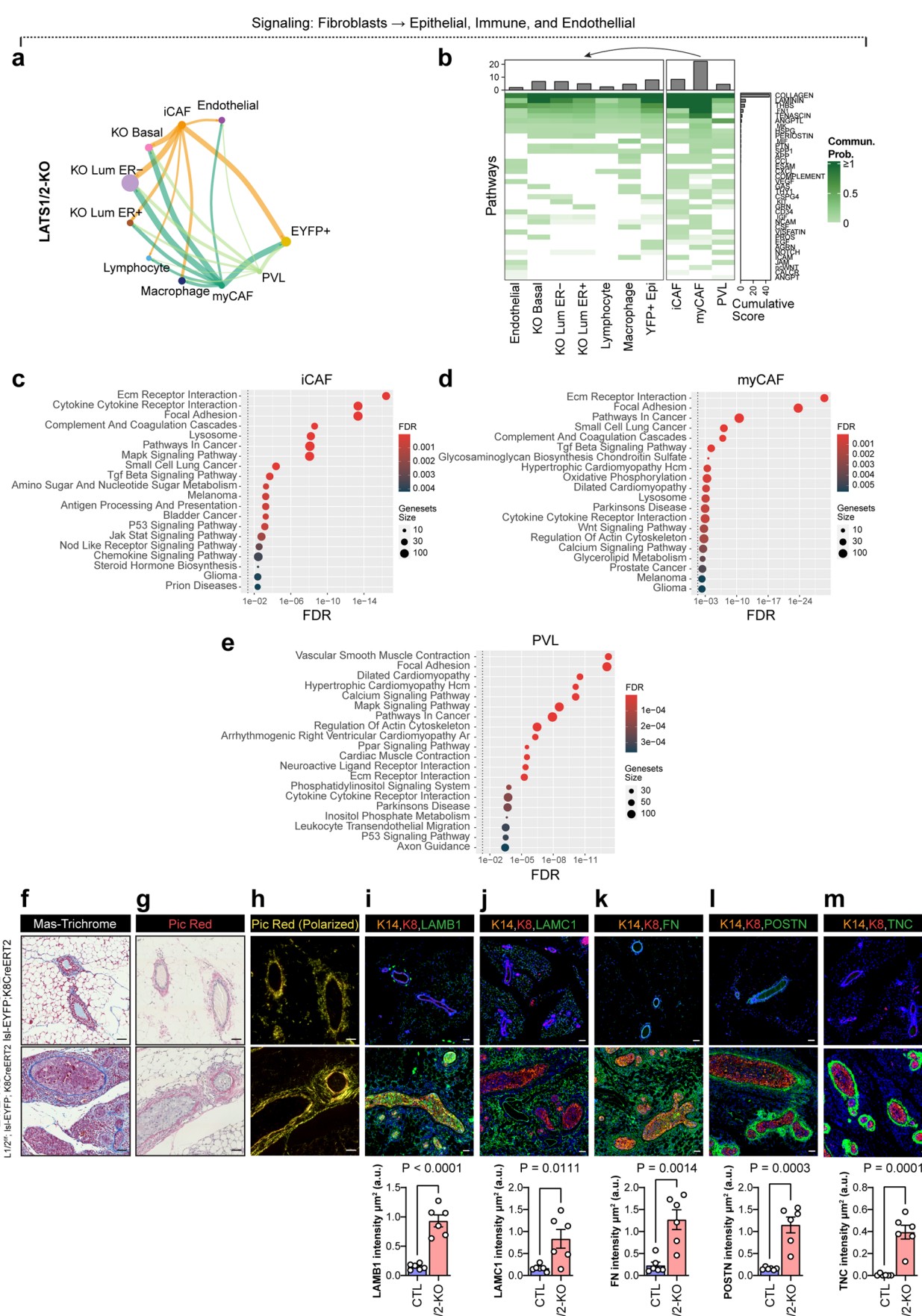

© The Author(s)

**Figure 4. Mammary carcinomas driven by LATS1/2 deletion display deposition of extracellular matrix proteins.**

(A) CellChat circle plots showing communication networks from iCAF, myCAF, and PVL groups (senders) to epithelial and stromal cell groups (receivers) in LATS1/2-KO mammary glands. Line widths represent the total number of interactions identified. (B) Heatmap of communication probabilities of signaling pathways from iCAF, myCAF, and PVL groups to epithelial and stromal cell groups in LATS1/2-KO mammary glands. (C–E) GSEA analysis of KEGG genesets enriched in the iCAF population (C), myCAF population (D), and PVL population (E) identified in mammary glands via single cell RNA-sequencing. (F) Masson-Trichrome staining of CTL and LATS1/2-KO mammary glands (Scale bar 50 μm) ($n = 6$ CTL mice, $n = 4$ L1/2-KO mice from two experiments). (G, H) Picrosirius red staining in CTL and LATS1/2-KO mammary glands using brightfield (G) and polarizing (H) light (Scale bars 50 μm) ($n = 4$ CTL mice, $n = 6$ L1/2-KO mice). (I–M) Immunofluorescence staining for KRT14, KRT8, and Laminin β1 (I), Laminin γ1 (J), Fibronectin (K), Periostin (L), and Tenascin C (M) in CTL and LATS1/2-KO mammary glands (Scale bars 50 μm) along with quantification of intensity/area for each ($n = 6$ CTL mice, $n = 6$ L1/2-KO mice, 5 regions analyzed per mouse for each panel (I–M). Data are shown with mean ± SEM. Two-tailed unpaired T-tests). Source data are available online for this figure.

in greater depth. We first turned to the transcriptional expression of ECM genes in our single-cell RNA-sequencing data. Enrichment analysis performed on the iCAF, myCAF, and PVL clusters with genesets from the Kyoto Encyclopedia of Genes and Genomes (KEGG) demonstrated enrichment of ECM-related genesets such as ECM Receptor Interaction and Focal Adhesion in our stromal populations, predominantly myCAFs and iCAFs (Fig. 4C–E). We next sought to validate changes in ECM proteins histologically in Lats1/2-null carcinomas. We observed a noticeable presence of collagen immediately surrounding the mammary ducts of Lats1/2ff; lsl-EYFP; Krt8CreERT2 mice through Masson-Trichrome and picrosirius red staining (Figs. 4F–H and EV3C–E). We noted the presence of collagen fibers intercalated within the stromal cells surrounding mammary ducts and developing DCIS lesions (Fig. 4F–H). We also examined expression of Laminin β1 (Figs. 4I and EV3F), Laminin γ1 (Figs. 4J and EV3G), Fibronectin (Figs. 4K and EV3H), Periostin (Figs. 4L and EV3I), and Tenascin C (Figs. 4M and EV3J), finding strong increases in expression for all of these in the stroma of Lats1/2ff; lsl-EYFP; Krt8CreERT2 mice relative to controls.

## Basal-like carcinomas driven by Lats1/2 inactivation mediate macrophage recruitment in the mammary stroma

The second most populous stromal cell type we observed in Lats1/2ff; lsl-EYFP; Krt8CreERT2 mice were immune cells in clusters 9 and 12 (Fig. 1E). Indeed, we observed a marked increase in CD45+ immune cells in the stroma of Lats1/2ff; lsl-EYFP; Krt8CreERT2 mice by immunofluorescence analysis (Fig. 5A). Our single cell data highlighted a clear increase in cells associated with cluster 9 in the Lats1/2ff; lsl-EYFP; Krt8CreERT2 mice (Fig. 5B), which expressed myeloid-related genes, with notably high expression of monocyte and macrophage-associated markers such as Cd68, Cd14, and Adgre1 (Fig. 5C). Histology for CD68, a surface marker that is predominantly expressed on monocytes and macrophages (Pulford et al, 1990), confirmed increases in these cells in the stroma of Lats1/2ff; lsl-EYFP; Krt8CreERT2 mice (Figs. 5D and EV4A). We further profiled these mice for F4/80, a marker specific for murine macrophages (McKnight et al, 1996), and found high expression of this throughout the mammary stroma and within the ducts and developing carcinomas (Figs. 5E and EV4B). Tumor-associated macrophages are associated with several oncogenic phenotypes in breast cancer, including TNBC (Qiu et al, 2022). To examine potential mechanisms of recruitment of macrophages, we performed a CellChat analysis designating the macrophage group as receiving cells. This revealed signals to macrophages from epithelial, CAF, lymphocyte, and endothelial

populations (Fig. 5F,G). Notable among these, the CSF pathway is a widely identified mediator of macrophage recruitment in tumors (Mantovani et al, 2022). We observed upregulation of Csf1 in Lats1/2-null cells relative to control cells via RNA-sequencing on EYFP+ cells sorted from the mammary glands of lsl-EYFP; Krt8CreERT2 and Lats1/2ff; lsl-EYFP; Krt8CreERT2 mice (Fig. 5H) (Kern et al, 2022). To confirm expression of Csf1 in the tissues of these mice, we performed RNAscope, which demonstrated strong increases in Csf1 expression predominantly in epithelial cells of Lats1/2-KO carcinomas (Figs. 5I and EV4C). Of note, CellChat analysis looking for signals being received by lymphocytes, which were represented in Cluster 12 in our scRNA-seq data, revealed that most signals originated from the myCAF and iCAF populations in the Lats1/2-KO samples (Fig. EV4D,E).

To define signals that originate from macrophages in Lats1/2-null carcinomas, we performed CellChat analysis designating macrophages as sender cells (Fig. 5J,K), which predicted several pathways as axes between macrophages and other cell types, with the strongest signals going to EYFP+ lineage-traced cells. Interestingly, many of these pathways were ECM-related, suggesting an additional role for macrophages in promoting the ECM-rich microenvironment we observe in Lats1/2ff; lsl-EYFP; Krt8CreERT2 mice.

## Reciprocal signaling between the stroma and epithelium associates with induced ECM-focal adhesion and YAP/TAZ responses throughout the Lats1/2-inactivated niche

Having shown changes to CAFs and macrophages in the stroma of Lats1/2ff; lsl-EYFP; Krt8CreERT2 mice, we next sought to examine how epithelial cells may respond to these stromal changes to elicit tumorigenesis. CellChat analysis identifying signals being received by epithelial groups in LATS1/2-KO samples showed that the highest number of signals are sent from CAFs, particularly myCAFs (Fig. 6A). When analyzing the pathways represented in these signals, the strongest pathways enriched were ECM-related pathways (Fig. 6B), suggesting that ECM-remodeling organizes key signaling axes that contribute to tumorigenic phenotypes observed in Lats1/2ff; lsl-EYFP; Krt8CreERT2 mice.

To define pathways activated in carcinoma cells in Lats1/2ff; lsl-EYFP; Krt8CreERT2 mice, we generated transcriptomic signatures from epithelial cells using our single-cell RNA-sequencing dataset and performed GSEA on genesets from KEGG. This revealed significant enrichment for genesets associated with focal adhesion assembly, ECM-receptor interactions, and actin cytoskeletal rearrangements (Fig. 6C). GSEA from prior bulk RNA-sequencing on sorted Lats1/2-null cells (Kern et al, 2022) likewise revealed enrichment for

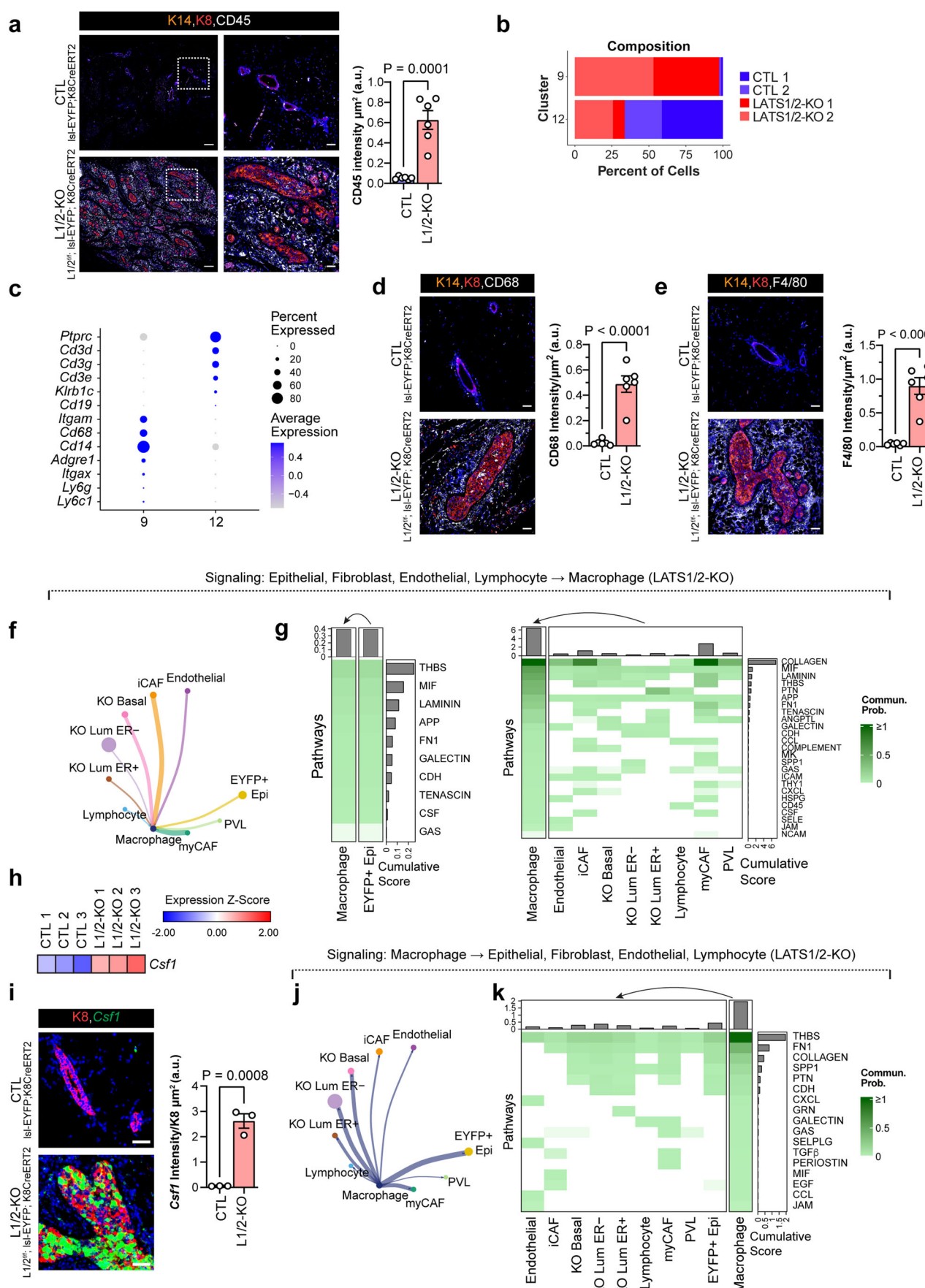

**Figure 5. Immune cell alterations and macrophage recruitment in carcinomas driven by LATS1/2 inactivation.**

(A) Immunofluorescence staining for KRT8, KRT14, and CD45 in CTL and LATS1/2-KO mammary glands and quantification of CD45 intensity/area. Shown on the right are subset images of the regions outlined by the dashed boxes on the left (Scale bar: 200 μm left images, 50 μm right images) ($n = 6$ CTL mice, $n = 6$ L1/2-KO mice. 5 regions analyzed per mouse. Data are shown with mean ± SEM. Two-tailed unpaired T-test. Experiment was repeated twice with similar results). (B) Composition of single-cell RNA-sequencing clusters 9 (monocytes/macrophages) and 12 (lymphocytes) colored by condition. (C) Dot heatmap of myeloid and lymphoid markers in the two immune cell clusters identified. (D) Immunofluorescence staining for KRT8, KRT14, and CD68 in CTL and LATS1/2-KO mammary glands and quantification of CD68 intensity/area. (Scale bar, 50 μm) ($n = 6$ CTL mice, $n = 6$ L1/2-KO mice. 5 regions analyzed per mouse. Data are shown with mean ± SEM. Two-tailed unpaired T-test. Experiment was repeated twice with similar results). (E) Immunofluorescence staining for KRT8, KRT14, and F4/80 in CTL and LATS1/2-KO mammary glands and quantification of F4/80 intensity/area (Scale bar, 50 μm) ($n = 6$ CTL mice, $n = 6$ L1/2-KO mice. 5 regions analyzed per mouse. Data are shown with mean ± SEM. Two-tailed unpaired T-test. Experiment was repeated twice with similar results). (F) CellChat circle plots showing communication networks from epithelial and stromal groups (senders) to macrophages (receivers) in LATS1/2-KO mammary glands. Line widths represent the total number of interactions identified. (G) Heatmap of communication probabilities of signaling pathways from epithelial and stromal cell groups to macrophages in LATS1/2-KO mammary glands. (H) Expression of *Csf1* in sorted EYFP$^+$ cells from the mammary glands of CTL (lsl-EYFP; Krt8CreERT2) and LATS1/2-KO (LATS1/2$^{ff}$; lsl-EYFP; Krt8CreERT2) mice. (I) RNAscope for *Csf1* along with IF for KRT8 in CTL and LATS1/2-KO mammary glands and quantification of *Csf1* intensity/area within K8+ epithelial cells (Scale bar, 50 μm) ($n = 3$ CTL mice, $n = 3$ L1/2-KO mice. 5 regions analyzed per mouse. Data are shown with mean ± SEM. Two-tailed unpaired T-test). (J) CellChat circle plots showing communication networks from macrophages (senders) to epithelial and stromal cells (receivers) in LATS1/2-KO mammary glands. Line widths represent the total number of interactions identified. (K) Heatmap of communication probabilities of signaling pathways from macrophages to epithelial and stromal cell groups in LATS1/2-KO mammary glands. Source data are available online for this figure.

genesets associated with cell-matrix interactions and focal adhesion signaling (Fig. EV4F). Histology analysis indicated increased expression of Integrin β1 in Lats1/2-KO carcinomas, with increased expression in both the epithelia and stroma of LATS1/2-KO mice relative to controls (Fig. 6D). Furthermore, treatment of LATS1/2-KO mammary organoid cultures (Kern et al, 2022) with PF-573228 a small molecule inhibitor of Focal Adhesion Kinase (FAK) (Slack-Davis et al, 2007), inhibited organoid growth (Fig. 6E). Together, these results point to interactions with the ECM and focal adhesion signaling as potential mediators of cellular phenotypes in our Lats1/2$^{ff}$; lsl-EYFP; Krt8CreERT2 model.

Considering epithelial, stromal, and immune cell types all associate with ECM remodeling in cancer (Winkler et al, 2020), the strong ECM phenotypes we observed suggested that microenvironment changes driven by Lats1/2 inactivation in luminal cells may initiate a niche allowing for oncogenic transformation of both epithelial cells and stromal cell populations, resulting in basal-like carcinomas. When analyzing enrichment of a previously curated list of conserved YAP/TAZ target genes in cancer (Wang et al, 2018) across all clusters in our single cell data, we observed the strongest enrichment in the clusters containing LATS1/2-KO epithelial cells, myCAFs, and iCAFs (Fig. 6F). The enrichment of YAP/TAZ signatures in these results suggested YAP/TAZ activation throughout the local microenvironment, including in a non-cell autonomous fashion within cells that were not LATS1/2-null. Indeed, histological analysis of TAZ revealed increased nuclear TAZ levels in both EYFP+ (Cre active) and EYFP-negative (Cre inactive) epithelial cells in Lats1/2$^{ff}$; lsl-EYFP; Krt8CreERT2 mice relative to luminal cells of control mice (Fig. 6G). YAP/TAZ exert their transcriptional activity through interactions with transcription factors in the nucleus, predominantly the TEAD family (TEAD1-4) (Pan, 2010). We found that the TEADs are expressed in a variety of cell types represented in our scRNA-seq experiment (Fig. EV4G) and that the clusters associated with LATS1/2-KO cells prominently expressed *Tead1*, with lower levels of *Tead2* and *Tead4*. Other epithelial cells expressed predominantly *Tead1* and/or *Tead2*, and myCAFs, PVL cells, and iCAFs most highly expressed *Tead1* relative to the other TEADs. To further explore relationships between TEADs and the cell populations identified in our scRNA-seq data we used SCENIC, a computational tool that combines single-cell gene expression with analysis of cis-regulatory motifs to predict transcription factors underlying cell states (Aibar et al, 2017). This analysis identified TEAD1 as among the top differentially expressed regulons associated with the epithelial program in LATS1/2-KO mammary glands (Fig. 6H). Notably, genes identified as part of the TEAD regulon in LATS1/2-KO-associated cells included signaling factors we demonstrated prior, including *Tgfb2*, *Pdgfa*, *Pdgfb*, and *Csf1* (Dataset EV2).

Lastly, to define whether inactivation of LATS1/2 and activity of YAP/TAZ/TEADs correlate with similar stromal phenotypes in human breast cancer, we used TIMER (Li et al, 2020), which is a computational platform that estimates stromal cell infiltration based on RNA-seq data from The Cancer Genome Atlas (TCGA). This analysis revealed a positive correlation between expression of the LATS1/2-KO epithelial signature and the presence of both CAF and macrophage populations in human breast tumors (Fig. EV4H). Together, these results point to TEAD activity as a potent orchestrator of stromal remodeling in breast tumors.

## Pharmacological inhibition of YAP/TAZ-TEAD activity reverses mammary epithelial and stromal phenotypes driven by Lats1/2 inactivation

Given that Lats1/2-inactivation resulted in a YAP/TAZ-activated microenvironment, we tested the consequence of inhibiting YAP/TAZ/TEAD activity by treating Lats1/2$^{ff}$; lsl-EYFP; Krt8CreERT2 mice with VT104, a small molecule inhibitor of TEAD palmitoylation and YAP/TAZ-driven oncogenesis (Tang et al, 2021). Inhibition of YAP/TAZ-TEAD activity showed a remarkable reduction in a variety of YAP/TAZ transcriptional targets in epithelial and stromal cells collected from the 3rd and 5th mammary glands of VT104-treated Lats1/2$^{ff}$; lsl-EYFP; Krt8CreERT2 mice relative to vehicle-treated mice (Fig. EV5A). Moreover, treatment of mice with VT104 led to reversal of carcinoma phenotypes in the Lats1/2$^{ff}$; lsl-EYFP; Krt8CreERT2 model, including a reversal of overgrowth of epithelial cells within the mammary ducts (Figs. 7A and EV5B) and the reversal of luminal-basal plasticity as assessed by K14 expression in EYFP+ cells (Figs. 7B and EV5C). We also noted reversal of stromal phenotypes, including a reduction in the accumulation of both PDGFRβ+ fibroblasts (Figs. 7C and EV5D) and F4/80+ macrophages around mammary ducts

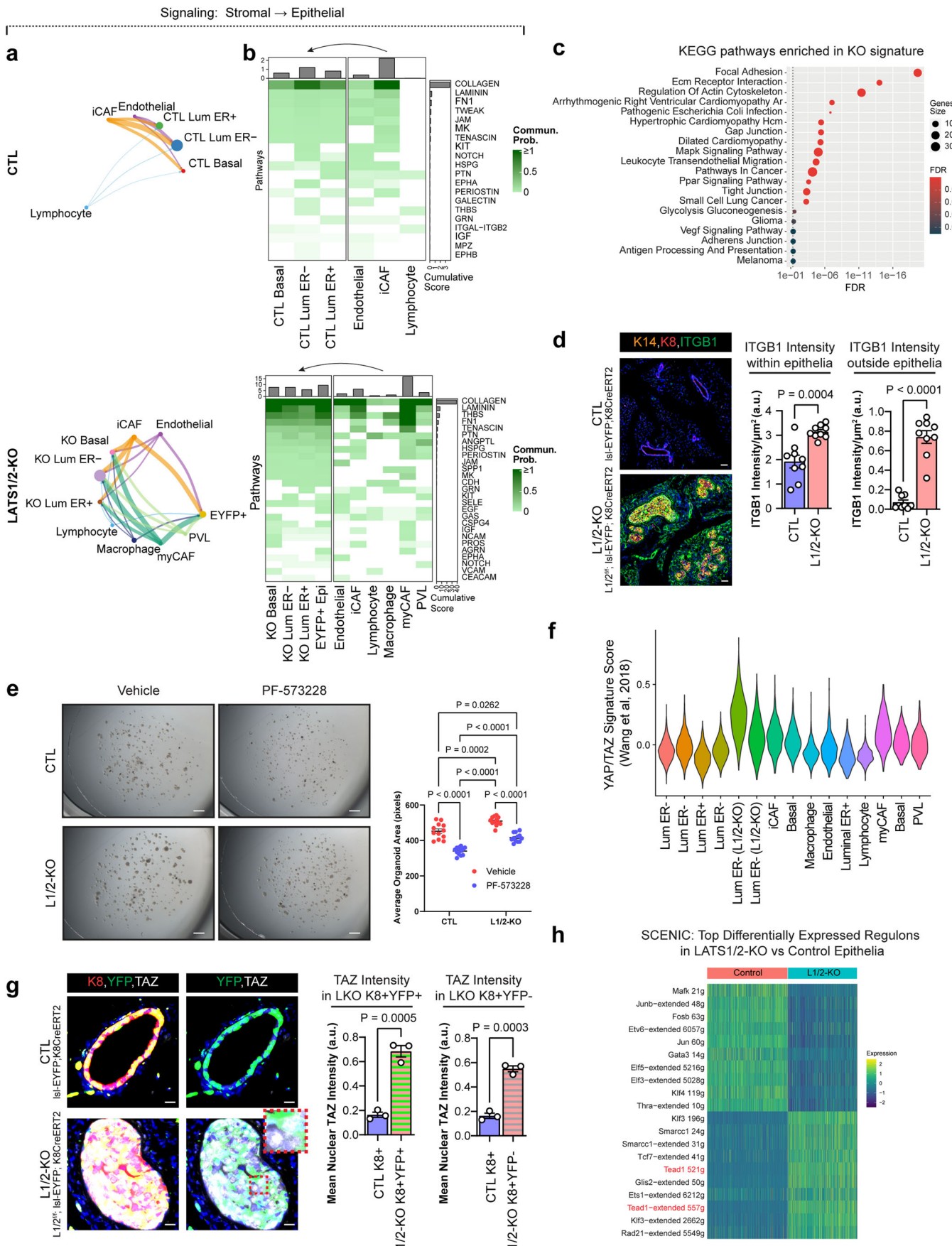

**Figure 6. Hippo inactivation in tumor cells drives a tumor-associated niche promoting YAP/TAZ activation.**

(A) CellChat circle plots showing communication networks from stromal groups (senders) to epithelial groups (receivers) in CTL and LATS1/2-KO mammary glands. Line widths represent the total number of interactions identified. (B) Heatmaps of communication probabilities of signaling pathways from stromal cell groups to epithelial cell groups in CTL and LATS1/2-KO mammary glands. (C) Enrichment analysis demonstrating enriched KEGG genesets in epithelial cell signatures from LATS1/2-KO mammary glands derived from single-cell RNA-sequencing data. (D) Immunofluorescence staining for KRT8, KRT14, and ITGB1 in CTL and LATS1/2-KO mammary glands, along with quantification of ITGB1 intensity/area within epithelia and outside of epithelia (Scale bar, 50 μm) ($n = 9$ CTL and 9 L1/2-KO mice pooled from two experiments. 5 regions analyzed per mouse. Data are shown with mean ± SEM. Two-tailed unpaired T-test). (E) Organoid cultures of control and LATS1/2-KO organoid cultures treated with vehicle (DMSO) or 5 μM PF-573228 (Scale bar, 1 mm), along with quantification of organoid size between conditions ($n = 12$ wells per condition. Data are shown with mean ± SEM. Ordinary two-way ANOVA with Tukey's multiple comparisons tests. Adjusted $P$-values are displayed in figure. Experiment was performed twice with 5 μM PF-573228 and once with 2.5 μM PF-573228, all with similar results). (F) Correlation of a previously identified cancer-associated YAP/TAZ signature (Wang et al, 2018) to all cell clusters identified from single-cell RNA-sequencing (from left to right: $n = 3496$ cells for C1, $n = 2666$ cells for C2, $n = 2458$ cells for C3, $n = 1607$ cells for C4, $n = 1483$ cells for C5, $n = 1156$ cells for C6, $n = 1031$ cells for C7, $n = 928$ cells for C8, $n = 799$ cells for C9, $n = 611$ cells for C10, $n = 603$ cells for C11, $n = 440$ cells for C12, $n = 417$ cells for C13, $n = 369$ cells for C14, $n = 140$ cells for C15). (G) Immunofluorescence staining for KRT8, YFP, and TAZ in CTL and LATS1/2-KO mammary glands and quantification of nuclear TAZ intensity in K8 + YFP+/− cells in LATS1/2-KO mice and K8+ cells in CTL mice. (Scale bar, 50 μm) ($n = 3$ CTL mice, $n = 3$ L1/2-KO mice. 11–14 regions analyzed per mouse. Data are shown with mean ± SEM. Two-tailed unpaired T-tests. Note that the same CTL samples were used for the comparisons in each plot). (H) SCENIC analysis identifying differentially expressed transcription factor regulons in epithelial cells from LATS1/2-KO mice and control mice. Source data are available online for this figure.

(Figs. 7D and EV5E). To test the necessity of YAP/TAZ/TEAD in directly regulating the expression of major signaling ligands we identified through our CellChat analyses prior, we performed RNAscope for the expression of *Tgfb2* (Fig. 7E), *Pdgfb* (Fig. 7F), and *Csf1* (Fig. 7G) in control and VT104-treated mice. These results demonstrated a robust reduction in the expression of *Tgfb2*, *Pdgfb*, and *Csf1* in the epithelia of VT104-treated LATS1/2-KO mice relative to vehicle-treated LATS1/2-KO controls. Thus, inhibition of YAP/TAZ-TEAD activity successfully reverses both epithelial and stromal phenotypes driven by Hippo pathway inactivation in the initiation of basal-like mammary carcinomas.

## Discussion

Considerable focus in the field of cancer research has been placed on the tumor microenvironment in recent years (de Visser and Joyce, 2023). The advent of single-cell RNA-sequencing has helped delineate distinct tumor-associated cell populations and has revealed the complexity of tumor ecosystems, including that of human breast tumors (Pal et al, 2021; Wu et al, 2021; Wu et al, 2020). However, many of these studies have been performed on invasive carcinomas and late-stage tumors, thus missing information on the signals required to establish a tumor-supportive niche in early oncogenesis. Moreover, knowledge into the signals that mediate this initiation has been lacking. The results we present here offer insight into these questions and provide new understanding of early events that promote the initiation of the basal-like breast cancer niche.

In this study we demonstrated that basal-like mammary carcinomas driven by Lats1/2 inactivation exhibit changes to epithelial, fibroblast, and immune cell populations. We used single-cell RNA-sequencing methodology and histology to identify separate cell types arising in basal-like carcinoma initiation and uncover phenotypic properties associated with these cells. Furthermore, we used computational prediction of ligand-receptor interactions between cells to demonstrate signaling patterns and pathways between these cell types. We identified specific stromal changes that occur in these carcinomas, including the expansion of distinct CAF populations, macrophage recruitment, and ECM

remodeling. These changes phenocopy observations in human TNBCs, which exhibit notable CAF populations, ECM remodeling (Acerbi et al, 2015; Kieffer et al, 2020; Wu et al, 2020) and recruitment of macrophages, which are becoming increasingly understood as mediators of tumor and immune cell phenotypes in breast cancer (Mehta et al, 2021; Qiu et al, 2022).

While much effort has been put into uncovering signaling mechanisms that mediate oncogenic properties intrinsic to tumor cells, identifying mechanisms that drive interactions between tumor and stromal cells is a less explored avenue. The Hippo pathway has traditionally been described as a mediator of cell-intrinsic properties in tumor cells that include the promotion of stem cell phenotypes and pro-growth and survival properties in tumor cells (Yu et al, 2015; Zanconato et al, 2016). Here we show that Hippo dysregulation through Lats1/2 deletion in the luminal mammary epithelium is sufficient to drive dramatic stromal remodeling resembling phenotypes observed in human breast cancers. Importantly, we demonstrate differential expression of many ligands involved in epithelial-stromal crosstalk in our carcinoma model, suggesting direct roles for YAP/TAZ activity in organizing the tumor niche. These include PDGF and TGF-β ligands, which were elevated in Lats1/2-null epithelia and predicted to communicate to CAF populations in developing tumors, including myCAFs, iCAFs and PVLs, suggesting that these signals play key roles in fibroblast activation and recruitment. Macrophage-recruiting ligands, such as *Csf1*, were also increased in Lats1/2-null epithelia, accompanied by the recruitment of myeloid cells that express distinct monocyte and macrophage-associated markers.

Most of the stromal cells that accumulated within the Lats1/2-null mammary carcinomas were enriched in the expression of ECM-related genes, including those encoding Collagens, Laminins, Thrombospondin, Fibronectin, and Tenascin. Indeed, Masson-Trichrome and picrosirius red staining showed high levels of collagens adjacent to the carcinomas and immunohistochemistry validated increased levels of Laminin β1, Laminin γ1, Fibronectin, Periostin and Tenascin C, most of which showed a discrete pattern adjacent to the carcinomas. This pattern was in-line with the finding that myCAFs express particularly high expression of ECM-related genes, as myCAFs showed a similar spatial pattern adjacent to the carcinomas. Comparable remodeling of the ECM is

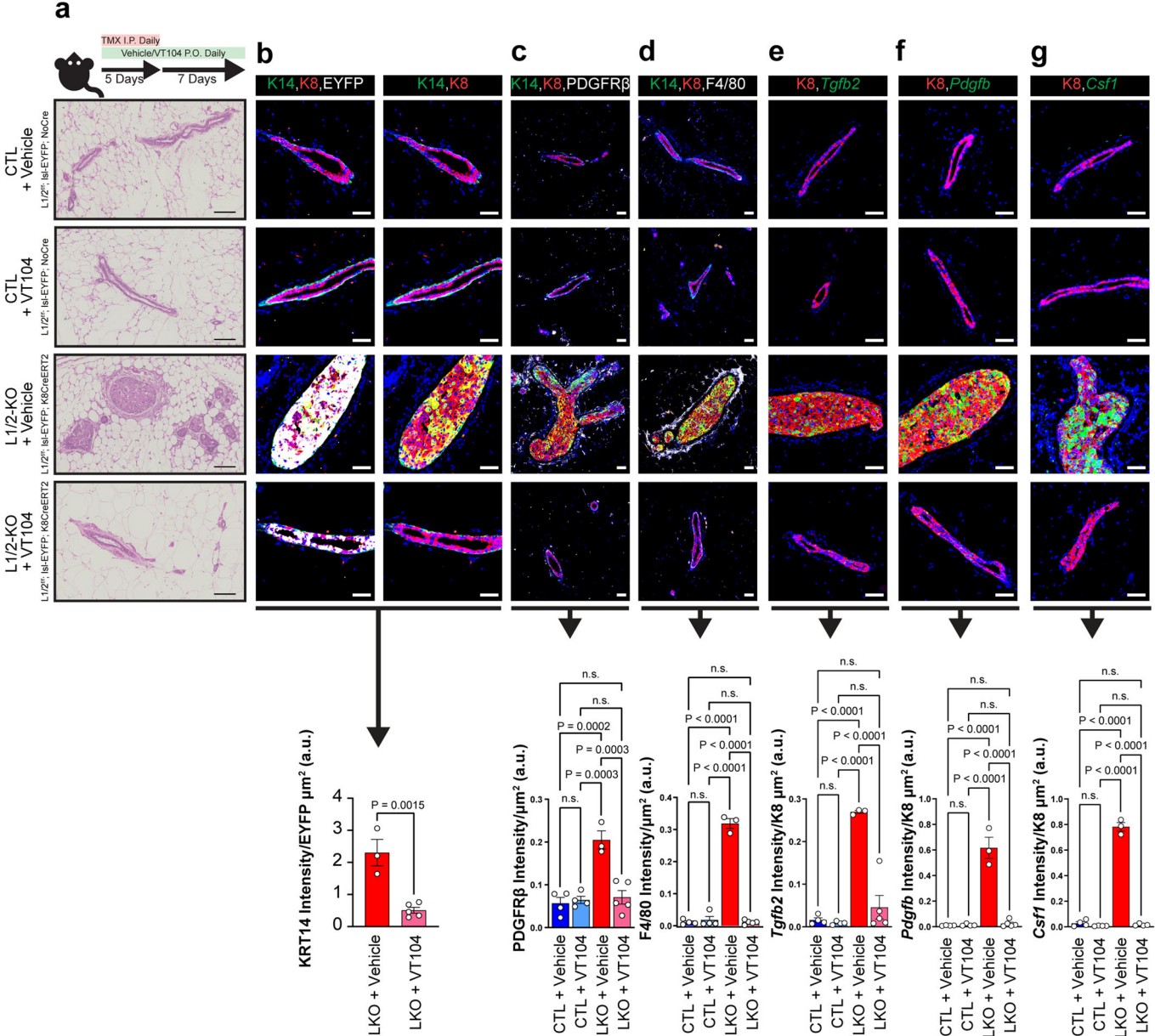

**Figure 7. Pharmacological inhibition of YAP/TAZ reverses epithelial and stromal alterations in LATS1/2-null carcinomas.**

(A) Treatment strategy for vehicle and VT104 (10 mg/kg) administration, along with hematoxylin and eosin stain of CTL (LATS1/2ff; lsl-EYFP; noCre) and LATS1/2-KO (LATS1/2ff; lsl-EYFP; Krt8CreERT2) mammary glands treated with Vehicle and VT104. (Scale bar, 100 μm) (n = 4 CTL + Vehicle mice, n = 4 CTL + VT104 mice, n = 3 L1/2-KO + Vehicle mice, n = 5 L1/2-KO + VT104 mice). (B) Immunofluorescence staining for KRT8, KRT14, and EYFP in mammary glands of LATS1/2-KO mice treated with Vehicle and VT104, along with quantification of K14 intensity/EYFP+ area (Scale bar, 50 μm) (n = 3 L1/2-KO + Vehicle mice, 5 L1/2-KO + VT104 mice. Data are shown with mean ± SEM. 5 regions analyzed per mouse. Two-tailed unpaired T-test). (C) Immunofluorescence staining for KRT14, KRT8, and PDGFRβ in mammary glands of CTL and LATS1/2-KO mice treated with Vehicle and VT104, along with quantification of PDGFRβ intensity/area (Scale bar, 50 μm). (D) Immunofluorescence staining for KRT14, KRT8, and F4/80 in mammary glands of CTL and LATS1/2-KO mice treated with Vehicle and VT104, along with quantification of F4/80 intensity/area (Scale bar, 50 μm). (E) RNAscope for Tgfb2 in mammary glands of CTL and LATS1/2-KO mice treated with Vehicle and VT104, along with IF for KRT8 and quantification of Tgfb2 intensity/K8+ area. (Scale bar, 50 μm). (F) RNAscope for Pdgfb in mammary glands of CTL and LATS1/2-KO mice treated with Vehicle and VT104, along with IF for KRT8 and quantification of Pdgfb intensity/K8+ area (Scale bar, 50 μm). (G) RNAscope for Csf1 in mammary glands of CTL and LATS1/2-KO mice treated with Vehicle and VT104, along with IF for KRT8 and quantification of Csf1 intensity/K8+ area. (Scale bar, 50 μm). For (C–F): n = 4 CTL + Vehicle mice, n = 4 CTL + VT104 mice, n = 3 L1/2-KO + Vehicle mice, n = 5 L1/2-KO + VT104 mice. For (G): n = 4 CTL + Vehicle mice, n = 4 CTL + VT104 mice, n = 3 L1/2-KO + Vehicle mice, n = 4 L1/2-KO + VT104 mice. 5 regions analyzed per mouse. Data are shown with mean ± SEM. Ordinary two-way ANOVAs with Tukey's multiple comparisons tests. Adjusted P-values are displayed in figure. Source data are available online for this figure.

well-documented in human TNBCs (Wu et al, 2021; Wu et al, 2020), and has been associated with carcinoma aggressiveness and therapy evasion (Kramer et al, 2019; van der Spek et al, 2020). Thus, our observations suggest that dysregulated Hippo pathway signaling contribute to the early stromal remodeling and ECM alterations that drive aggressive features in human breast cancer.

Interestingly, our data suggest that the stromal-ECM remodeling initiated by Lats1/2-inactivation creates a microenvironment that enables non-cell autonomous activation of YAP/TAZ. This conclusion is supported by observed increases in TAZ nuclear localization in both EYFP+ (Cre active; Lats1/2-null) and EYFP-negative (Cre inactive) epithelium. Elevated Integrin β1 was observed in the mammary epithelium and stroma of Lats1/2$^{ff}$; lsl-EYFP; Krt8CreERT2 mice. As Integrin β1 is a key receptor known to transmit signals from the ECM to cellular phenotypes, this offers a potential signal promoting the non-autonomous activation of YAP/TAZ (Su et al, 2024). The collective elevation of YAP/TAZ activity prompted us to consider newly emerging small molecule inhibitors as a potential therapeutic avenue for Lats1/2-null mammary phenotypes. Inhibitors of the TEAD transcription factors have shown effectiveness in inhibiting the growth of YAP/TAZ activated tumor cells (e.g., Nf2-mutated mesothelioma cells) and are currently being tested in cancer clinical trials. The TEAD transcription factors are also expressed across many cell populations in our scRNA-seq data, with *Tead1* predominantly expressed in the Lats1/2-null epithelium, along with lower levels of *Tead2* and *Tead4*. Treatment of Lats1/2$^{ff}$; lsl-EYFP; Krt8CreERT2 mice with VT104, a pan-TEAD inhibitor that blocks the palmitoylation pocket of TEADs (Tang et al, 2021), reversed all carcinoma phenotypes, including blocking the development of the tumor-stroma niche. These observations highlight TEAD inhibitors as a potential treatment for basal-like breast cancers that could allow for effective eradication of tumor cells and inhibition of changes to the stroma that create a pro-tumorigenic microenvironment.

# Methods

## Mice

Animal experiments in this study were carried out under protocols approved by the Boston University IACUC (Protocol # PROTO201800389). The mouse strains used are listed in the Reagents and Tools Table. Adult nulliparous animals were used for all experiments in this study and were between 3 and 10 months old at the initiation of experiments. Animals were housed under conditions of 68–79 °C, 30–70% humidity, and in facilities with 14/10 h and 12/12 h light/dark cycles. For induction of Cre activity, mice were intraperitoneally administered 2 mg of Tamoxifen (Sigma, T5648) on 5 consecutive days. The Tamoxifen was diluted in corn oil (Sigma, C8267). For VT104 treatment, mice were administered either vehicle or VT104 (10 mg/kg) once daily via oral gavage. The vehicle used was 5% DMSO (Fisher, D136-1 or D128-5), 5% Tween-80 (Millipore-Sigma, P1754), 40% PEG-400 (Electron Microscopy Sciences, #19720), and 50% PBS (Corning, 21-040-CV). To randomize mice for drug treatments, mice were first grouped to best normalize for differences in age, litter, and weight, and these groups were then randomly assigned to a treatment group (vehicle or VT104).

## Sample preparation for single-cell RNA-sequencing

The third, fourth, and fifth pairs of mammary glands from each mouse were excised and lymph nodes removed from the fourth pair. Following this, each sample was chopped using a McIlwain tissue chopper (Ted Pella). Samples were then dissociated using a solution of 2 mg/mL collagenase A (Roche, 11088793001), 0.5 units/mL dispase (Corning, 354235), 0.01 mg/mL DNAse (Roche, 4716728001), and 1x penicillin-streptomycin (Gibco, 30-002-CI) in DMEM (Corning, 10-103-CV) with rotating at 37 °C for 1 h. ACK Lysing Buffer (Gibco, A1049201) was then used to lyse red blood cells, TrypLE Express (Gibco, 12604021) with 0.01 mg/mL DNAse was used to dissociate samples to single cells, and samples were strained using a 40 μM cell strainer. Samples were then incubated in 1 μM Calcein Blue, AM (Invitrogen, C1429) for 15 min, washed with PBS, and strained into tubes for flow cytometry. Live cells were sorted into a solution of 0.04% BSA in PBS to proceed with processing for single-cell sequencing.

## Single-cell RNA-sequencing quality control and clustering

Cells with more than 500 features detected per cell were loaded to Seurat Version 4.1.3 (Hao et al, 2021), and cells with <20% reads coming from mitochondrial genes were retained for a total of 47,209 genes across 18,204 cells. Raw counts were normalized with Seurat's SCTransform method, and clustered using the top 20 PCAs. Cells were first classified using the SingleR algorithm, and then refined using known cell markers. Clustering was performed at a resolution of 0.4.

## Single-cell RNA-sequencing data analysis

UMAPs, dot heatmaps, violin plots, and differential expression heatmaps were generated using Seurat Version 4 (Hao et al, 2021). For the comparison iCAF, myCAF, and PVL signatures to those observed in human TNBC and generation of the associated violin plots (Fig. 3D) signatures for iCAFs, myCAFs, dPVLs, and imPVLs were first obtained from Wu et al (2020). Seurat's AddModuleScore was then used to calculate a signature score for each group per cell. Briefly, AddModuleScore determines the average expression of the given set of genes and subtracts the average expression of a control set of genes that have similar expression levels as the signature set for each cell. A similar process was used to compare the YAP/TAZ cancer signature (Wang et al, 2018) to single-cell clusters in Fig. 6F. Heatmaps with top differentially-expressed genes between clusters were generated using the FindAllMarkers function in Seurat. The analysis in Fig. 3C was performed only on the three fibroblast clusters after taking them as a subset of the Seurat object.

For CellChat analyses, the single-cell sequencing dataset was split into two Seurat objects, one containing cells from the control mice (n = 2), and the other containing cells from the Lats1/2-KO mice (n = 2). The SCT normalized data was used as input to CellChat for both objects, and only clusters containing at least 30 cells across the two replicates in each object were considered for signaling. This excluded two fibroblast/stromal groups (C13 and C15) and macrophages (C10) from analyses on the control samples due to low cell abundance. Communication Probabilities were

**Reagents and tools table**

| Reagent/Resource | Reference or Source | Identifier or Catalog Number |
|---|---|---|
| **Experimental models** | | |
| Lats1tm1.1Jfm/RjoJ | The Jackson Laboratory | #024941 |
| Lats2tm1.1Jfm/RjoJ | The Jackson Laboratory | #025428 |
| STOCK Tg(Krt8-cre/ERT2)17Blpn/J | The Jackson Laboratory | #017947 |
| B6.129×1-Gt(ROSA)26Sortm1(EYFP)Cos/J | The Jackson Laboratory | #006148 |
| **Antibodies** | | |
| Keratin 8 | DSHB | TROMA-1c |
| Keratin 14 – FITC | Millipore Sigma | CBL197F |
| PDGFRβ | Cell Signaling Technology | 3169 |
| CD45 | Cell Signaling Technology | 70257 |
| CD68 | Cell Signaling Technology | 97778 |
| F4/80 | Cell Signaling Technology | 70076 |
| αSMA-Cy3 | Millipore Sigma | C6198 |
| Laminin beta-1 | Invitrogen | MA5-32577 |
| Laminin gamma-1 | Abcam | Ab233389 |
| Fibronectin | Novus Biologicals | NBP1-91258 |
| Periostin | Abcam | Ab14041 |
| Tenascin C | Abcam | Ab108930 |
| GFP | Aves Labs | GFP-1020 |
| TAZ | Cell Signaling Technology | 83669 |
| Integrin β1 | Cell Signaling Technology | 34971 |
| Donkey α-Rabbit AF647 | Jackson ImmunoResearch | 711-606-152 |
| Donkey α-Rat AF488 | Jackson ImmunoResearch | 712-546-153 |
| Donkey α-Rat AF488 | Invitrogen | A48269 |
| Donkey α-Rat Cy3 | Jackson ImmunoResearch | 712-165-153 |
| Donkey α-Chicken AF594 | Jackson ImmunoResearch | 703-585-155 |
| Donkey α-Chicken AF647 | Jackson ImmunoResearch | 703-605-155 |
| **Oligonucleotides for RT-qPCR** | Forward Primer | Reverse Primer |
| *Ppia* (mouse) | GAGCTGTTTGCAGACAAAGTTC | CCCTGGCACATGAATCCTGG |
| *Ctgf* (mouse) | AGACCTGTGGGATGGGCAT | GCTTGGCGATTTTAGGTGTCC |
| *Cyr61* (mouse) | TAAGGTCTGCGCTAAACAACTC | CAGATCCCTTTCAGAGCGGT |
| *Ankrd1* (mouse) | TGCGATGAGTATAAACGGACG | GTGGATTCAAGCATATCTCGGAA |
| *Tgfb2* (mouse) | GCGGACGATTCTGAAGTAGG | GGAGTACTACGCCAAGGAGGT |
| *Serpine1* (mouse) | GTAGCACAGGCACTGCAAAA | GCCGAACCACAAAGAGAAAG |
| *Birc5* (mouse) | GACTGCAAAGACTACCCGTCA | GATGTGGCATGTCACTCAGG |
| **Chemicals, Enzymes and other reagents** | | |
| Tamoxifen | Sigma | T5648 |
| Corn Oil | Sigma | C8267 |
| VT104 | Sigma | SML3445 |
| DMSO | Fisher | D136-1 and D128-5 |
| Tween-80 | Millipore-Sigma | P1754 |
| PEG-400 | Electron Microscopy Sciences | 19720 |
| PBS | Corning | 21-040-CV |
| Collagenase A | Roche | 11088793001 |

| Reagent/Resource | Reference or Source | Identifier or Catalog Number |
|---|---|---|
| Dispase | Corning | 354235 |
| DNAse | Roche | 4716728001 |
| Penicillin-streptomycin | Gibco | 30-002-CI |
| DMEM | Corning | 10-103-CV |
| ACK Lysing Buffer | Gibco | A1049201 |
| TrypLE Express | Gibco | 12604021 |
| Calcein Blue, AM | Invitrogen | C1429 |
| Paraformaldehyde | Electron Microscopy Sciences | 15710 |
| Paraformaldehyde | Santa Cruz Biotechnology | SC281692 |
| Paraffin | Leica Microsystems | 39602004 |
| Hematoxylin | Epredia | 6765003 |
| Eosin | Epredia | 71311 |
| Cytoseal XYL | Epredia | 83124 |
| Antigen Unmasking Solution, Citrate-Based | Vector Laboratories | H3300 |
| Donkey Serum | Millipore Sigma | S30 |
| Rodent Block M | Biocare Medical | RBM961 |
| Prolong Gold Antifade Mountant with DAPI | Invitrogen | P36931 |
| RNAscope 2.5 HD-RED assay | Advanced Cell Diagnostics | 322360 |
| RNAscope Multiplex Fluorescent v2 assay | Advanced Cell Diagnostics | 323370 |
| Mm-Tgfb2 | Advanced Cell Diagnostics | 406181 |
| Mm-Pdgfa | Advanced Cell Diagnostics | 411361 |
| Mm-Pdgfb | Advanced Cell Diagnostics | 424651 |
| Mm-Csf1 | Advanced Cell Diagnostics | 315621 |
| Positive Control Probe - Mm-Ppib | Advanced Cell Diagnostics | 313911 |
| Negative Control Probe - DapB | Advanced Cell Diagnostics | 310043 |
| Mm-Scara5-O1 | Advanced Cell Diagnostics | 522301 |
| Mm-Lrrc15-C2 | Advanced Cell Diagnostics | 467831-C2 |
| Mm-Notch3-C3 | Advanced Cell Diagnostics | 425171-C3 |
| 3-plex Positive Control Probe | Advanced Cell Diagnostics | 320881 |
| 3-plex Negative Control Probe | Advanced Cell Diagnostics | 320871 |
| Trichrome Stain (Masson) Kit | Millipore Sigma | HT15 |
| Weigert's Iron Hematoxylin | Sigma | HT1079 |
| Phosphomolybdic acid | Sigma | HT153 |
| 0.1% Sirius Red in saturated picric acid | Electron Microscopy Sciences | 2635702 |
| Hydrochloric acid | Honeywell | 258148 |
| RNEasy Mini Kit | Qiagen | 74106 |
| iScript cDNA Synthesis Kit | BioRad | 1708891 |
| Fast SYBR Green Master Mix | Applied Biosystems | 4385612 |
| Matrigel Basement Membrane Matrix | Corning | 356237 |
| DMEM-F12, no glutamine | Thermo-Fisher | 21331020 |
| Insulin-Transferrin-Selenium-Ethanolamine | Thermo-Fisher | 51500056 |
| Epidermal Growth Factor (EGF), Mouse Natural | Corning | CB-40001 |
| 4-Hydroxytamoxifen | Sigma-Aldrich | H7904 |
| PF-573228 | SelleckChem | S2013 |

| Reagent/Resource | Reference or Source | Identifier or Catalog Number |
|---|---|---|
| **Software** | | |
| Seurat v4.1.3 | Hao et al, 2021 | |
| GSVA | Hanzelmann et al, 2013 | |
| SCENIC v 1.3.1 | Aibar et al, 2017 | |
| ZEN 3.1 Blue | Zeiss | |
| ZEN 3.3 Blue | Zeiss | |
| QuantStudio Real-Time PCR Software v1.6.1 | Applied Biosystems | |
| ZEN 2 Blue | Zeiss | |
| CellProfiler 4 | Stirling et al, 2021 | |
| **Other** | | |
| Microtome | Leica | RM2235 |
| Tissue chopper | Ted Pella | |
| Decloaking chamber | Biocare Medical | DC2012 |
| Axio Scan.Z1 | Zeiss | |
| Axio Observer.D1 | Zeiss | |
| ViiA 7 Real-Time PCR System | Applied Biosystems | |
| SteREO Discovery.V12 | Zeiss | |

calculated using all other default parameters. To define the EYFP$^+$ group, cells with normalized expression value of *EYFP* > 0.5, and falling within the KO Lum ER-, KO Lum ER+, or KO Basal clusters were selected for analyses and marked as EYFP$^+$ Epithelial cells.

To derive transcriptomic signatures for KO epithelial cells in the single-cell RNA-sequencing dataset, the FindMarkers function in Seurat was used to identify a set of genes unregulated in the KO samples when compared to the control samples. Signatures for each condition were defined if they had a log2FC >=0.6 and adjusted *p* value <=0.05. For enrichment analysis performed on these signatures, hypeR was used to perform gene set enrichment analysis using a background of 47,000 genes using the KEGG database.

For TIMER analysis, TCGA BrCa samples were scored with the KO and CTRL epithelial cell signatures derived from the single cell using GSVA (Hanzelmann et al, 2013). A combined KO-CTRL score was made by subtracting the CTRL GSVA signature score from the KO GSVA signature score for each BrCa sample. TIMER (Li et al, 2020) deconvolution for BrCa tumor samples were downloaded from http://timer.cistrome.org/, which contains deconvolution scores for a number of cell types based on six databases: TIMER (Li et al, 2020), CIBERSORT (Newman et al, 2015), quanTIseq (Finotello et al, 2019), xCell (Aran et al, 2017), MCP-counter (Becht et al, 2016), and EPIC (Racle et al, 2017). For BrCa tumor samples that had deconvolution scores, purity scores, and expression values (N = 1045), the association between tumor sample deconvolution score and the KO-CTRL Epithelial Signature was assessed by Pearson correlation. For all datasets except EPIC and quanTIseq, which used direct correlations, a partial correlation conditioned on the purity score was performed. An fdr adjustment was performed across cell types within each database.

SCENIC v 1.3.1 (Aibar et al, 2017) was used to perform a regulon analysis on all epithelial clusters (CTL Lum ER-, CTL Lum ER+, KO

Lum ER-, CTL Basal, KO Basal, KO Lum ER+). The expression matrix containing raw counts was used as input to SCENIC with the motif collection version 9 containing 24k motifs. The matrix was filtered to remove genes that are expressed either at very low levels (< a UMI of 3 in 1% of the cells) or in too few cells (<1% of cells). GENIE3 was used to build the coexpression network on the log-transformed gene matrix and build and score the GRN. The AUC score per regulon per cell was added to the seurat object, and a differential regulon analysis was performed using the AUC scores with Seurat's FindMarkers, utilizing a Wilcoxon rank sum test.

## Immunofluorescence, immunohistochemistry, and RNAscope in situ hybridization

The fourth mammary glands were harvested from animals and fixed in 4% paraformaldehyde (Electron Microscopy Sciences, # 15710 or Santa Cruz Biotechnology SC281692) overnight, then dehydrated and embedded in paraffin (Leica Microsystems 39602004). Tissues were sectioned using a microtome (Leica RM2235) and charged microscopy slides (Epredia and Fisherbrand). For hematoxylin and eosin staining, tissue sections were deparaffinized and hydrated, incubated in hematoxylin (Epredia, 6765003) for seven minutes, washed under running water for ten minutes, differentiated by dipping in 1% acid alcohol, and blued by placing in Scott's tap water for three minutes. Slides were counterstained with eosin (Epredia 71311) for two minutes, followed by dehydration and mounting with Cytoseal XYL (Epredia, 83124).

For immunofluorescence staining, tissue sections were deparaffinized and hydrated, followed by antigen retrieval using either a decloaking chamber (Biocare Medical, DC2012) or microwave using a citrate-based antigen retrieval buffer (Vector Laboratories, H3300). Samples were blocked using 5% donkey serum (Millipore Sigma, S30) in TBS-T for one hour at room temperature.

αSMA-Cy3 staining was additionally preceded with a block using Rodent Block M (Biocare Medical, RBM961) for 30 min prior to blocking with 5% donkey serum in TBS-T. Following this, tissues were incubated overnight at 4–5 °C in the desired primary antibodies diluted in blocking buffer. Slides were then washed with TBS-T and secondary antibodies in blocking buffer were applied for one hour at room temperature. Following this, slides were washed with TBS-T and mounted using Prolong Gold reagent with DAPI (Invitrogen, P36931). DAPI is shown in blue in all figure panels. Antibodies used for immunofluorescence are described in the Reagents and Tools Table and used at the following dilutions: Keratin 8, ~0.5 μg/mL; Keratin 14 – FITC, 1:20; PDGFRβ, 1:100; CD45, 1:400; CD68, 1:400; F4/80, 1:400; αSMA-Cy3, 1:400; Laminin beta-1, 1:100; Laminin gamma-1, 1:100; Fibronectin, 1:200; Periostin, 1:200; Tenascin C, 1:200; GFP, 1:400; TAZ, 1:100; Integrin β1, 1:100; Donkey α-Rabbit 647, 2.5 μg/mL; Donkey α-Rat 488 (Jackson ImmunoResearch), 2–2.5 μg/mL; Donkey α-Rat 488 (Invitrogen), 4 μg/mL; Donkey α-Rat Cy3, 2.5 μg/mL; Donkey α-Chicken 594, 2.5 μg/mL; Donkey α-Chicken 647, 2.5 μg/mL.

RNAscope in situ hybridization was performed using either the RNAscope 2.5 HD-RED assay (Advanced Cell Diagnostics, 322360) or the Multiplex Fluorescent v2 assay (Advanced Cell Diagnostics, 323370) following the manufacturer's protocols with the following changes: for both assays, hydrogen peroxide incubation was performed prior to antigen retrieval, antigen retrieval was performed using a decloaking chamber, and protease incubation was performed for 15 min; for the 2.5 HD-RED Assay, the AMP 5 step was performed for 15–20 min. When RNAscope was coupled with immunofluorescence, slides were washed after completion of the RNAscope assay and immediately blocked with 5% donkey serum, followed by immunofluorescent staining as described above. For all RNAscope experiments, a negative control probe was run as a quality control. The probes used were obtained from Advanced Cell Diagnostics as follows: Mm-Tgfb2 (406181), Mm-Pdgfa (411361), Mm-Pdgfb (424651), Mm-Csf1 (315621), Positive Control Probe - Mm-Ppib (313911), Negative Control Probe - DapB (310043), Mm-Scara5-O1 (522301), Mm-Lrrc15-C2 (467831-C2), Mm-Notch3-C3 (425171-C3), 3-plex Positive Control Probe (320881), 3-plex Negative Control Probe (320871).

Masson-Trichrome staining was performed with a commercial kit (Millipore Sigma, HT15-1KT) using a staining protocol adapted from the standard manufacturer's microwave procedure and using Weigert's iron hematoxylin to stain nuclei. For picrosirius red staining, deparaffinized and rehydrated slides were stained with Weigert's iron hematoxylin (Sigma, HT1079) for 5 min, then rinsed under running water for 10 min. The slides were then treated with 0.2% phosphomolybdic acid (Sigma, HT153) for 5 min followed by staining with 0.1% Sirius Red in saturated picric acid (Electron Microscopy Sciences, 2635702) for 90 min. This was followed by a 2-minute wash in 0.01 N hydrochloric acid (Honeywell, 258148), a one-minute rinse with 70% ethanol, dehydration, and mounting using Cytoseal XYL (Epredia, 83124). Histology images were acquired using a Zeiss Axio Scan.Z1 microscope with ZEN 3.1 Blue software or a Zeiss Axio Observer.D1 microscope with ZEN 3.3 Blue software.

## RNA isolation and RT-qPCR

The 3rd and 5th mammary glands were excised from mice and chopped using a McIlwain tissue chopper (Ted Pella). Samples were then dissociated using the same buffer as described for sample preparation for single-cell RNA-sequencing. ACK Lysing Buffer (Gibco, A1049201) was used to lyse red blood cells, samples were washed once with PBS, centrifuged, and the resulting pellet of epithelial and stromal cells was lysed using RLT buffer from the RNEasy Mini Kit (Qiagen, 74106). RNA isolation was then performed using this kit. cDNA synthesis was performed using the iScript cDNA synthesis kit (BioRad, 1708891), and RT-qPCR analysis was performed using Fast SYBR Green reagents (Applied Biosystems, 4385612) on the ViiA 7 platform using QuantStudio Real-Time PCR Software v1.6.1 (Applied Biosystems). Primers used for RT-qPCR are listed in the Reagents and Tools Table. *Ppia* was used as a reference gene and fold change values for RT-qPCR data were obtained using the $2^{-ddCT}$ method. Fold change values were Log2-transformed for presentation and statistical analysis.

## Mammary organoid cultures

The 3rd, 4th, and 5th mammary glands were collected and dissociated for one hour rotating at 37 °C using the same solution used for sample preparation for single-cell RNA-sequencing. Adipocytes were eliminated using centrifugation and red blood cells were lysed using ACK lysing buffer (Gibco, A1049201). The epithelial fraction was further separated from stromal cells using two methods. First, differential centrifugation was performed using ten quick pulses in a centrifuge with media aspiration and resuspension between each. Second, differential adhesion was performed by plating the epithelial and any residual stromal cells on tissue culture plates for 1–1.5 h to allow stromal cells to adhere. The remaining epithelial organoid fraction was then collected and seeded in Matrigel domes (Corning, 356237) on round coverslips in 24-well plates. Organoids were counted to normalize numbers seeded between conditions. Organoids were grown in DMEM/F12 media without glutamine (Thermo-Fisher, 21331020) supplemented with 1x penicillin-streptomycin, 1x Insulin-Transferrin-Selenium-Ethanolamine (Thermo-Fisher 51500056), and 2.5 nM mouse EGF (Corning, CB-40001). For each experiment, two mice were pooled per condition (control or LATS1/2-KO). Organoids were grown for seven days, switched to media with 0.1 μM 4-Hydroxytamoxifen (Sigma-Aldrich, H7904) and either vehicle or PF-573228 (SelleckChem, S2013), and grown seven more days prior to imaging. Imaging of organoids was performed on a Zeiss SteREO Discovery.V12 microscope with ZEN 2 Blue software.

## RNA-sequencing expression analysis

Heatmaps for expression of RNA-sequencing data were generated using Morpheus software from the Broad Institute (https://software.broadinstitute.org/morpheus).

## Image analysis

All quantifications were performed in CellProfiler 4 (Stirling et al, 2021). For all quantifications on mouse tissues, five representative image subsets containing mammary ducts were captured for each mouse using KRT14 and/or KRT8 stain as a reference and while remaining blinded to the image channel to be quantified. For quantifications in Figs. 3A, 4I, 4J, 4K, 4L, 4M, 5A, 5D, 5E, 7C, and D, the subset images were then loaded into CellProfiler and the *MeasureImageIntensity* function was used to determine the total

intensity of the image channel to be quantified. These values were then divided by the scaled size of the image (in µm) to obtain intensity/area values per image, and the intensity/area values of all images for each mouse were averaged to obtain the data presented.

For quantifications in Figs. 3J, 3K, 3L, 5I, 7E, 7F, and 7G, the *Threshold* function in CellProfiler was first used to threshold the image subsets using the K8 stain. K8-thresholded images were then used to mask the channel to be quantified, and *MeasureImage Intensity* was used to calculate the total intensity of the image channel to be quantified within the masked K8 regions. *MeasureImageAreaOccupied* was then used to determine the area of the masked K8 region, and these values were converted to µm. The intensity values were then divided by the area values to obtain intensity/area values per image, and the intensity/area values of all images for each mouse were averaged to obtain the data presented. The measurements in Fig. 6D were obtained the same way, except K8 and K14 were used to threshold epithelia instead of K8 alone, and the invert of the K8/K14 mask was also used to calculate the intensity of the image channel outside of K8/K14+ regions. For Fig. 7B, images were thresholded on EYFP using the *Threshold* function, and then the K14 channel was masked by the EYFP-thresholded images. The total intensity of K14 within the EYFP mask was then measured using *MeasureImageIntensity*. The K14 intensity values were then divided by the area of the EYFP threshold in µm to obtain the data presented.

For quantification of TAZ intensity in EYFP+ and EYFP- cells, images were taken of ducts with a mixture of EYFP+ and EYFP- cells while blinded to TAZ staining. Images were then sequentially thresholded on K8 to identify epithelia and DAPI to identify nuclei using the *Threshold* function. For Lats1/2-KO samples, images were additionally thresholded on EYFP to identify EYFP+ and EYFP- cells. The K8-thresholded images were then masked using EYFP and the invert of EYFP to give K8 + EYFP+ and K8 + EYFP- cells. The DAPI threshold images were then masked by the resultant K8 + EYFP+ and K8 + EYFP- images to identify nuclei. Finally, the resultant masked K8 + EYFP+ nuclei and K8 + EYFP- nuclei were used to mask TAZ, and the mean nuclear TAZ intensity was calculated across all images taken per mouse using *MeasureImageIntensity*. The CTL mice were analyzed using the same settings, but without including an EYFP threshold, thus giving only K8+ nuclear TAZ intensity. For this experiment, nuclear brightness of the images was adjusted between mice to account for variability and ensure accurate identification of nuclei in the CellProfiler pipeline.

For the quantification in Fig. EV1A, the *Threshold* function in CellProfiler was first used to threshold the image subsets using the K8 and K14 stain. Nuclei were then identified as primary objects using DAPI, and the K8/14-thresholded images were used to mask the nuclei objects. The number of objects (nuclei) in the inverse region of the K8/K14-thresholded images were then counted in CellProfiler. The area of the K8/K14-inverse region was calculated in µm², and the number of stromal nuclei/µm² was then calculated. For all image data analyses, "a.u." = arbitrary units. Tissues from mice were excluded from image analyses if they had poor histological quality.

For the organoid size quantification in Fig. 6E, brightfield pictures of organoid wells were first loaded into CellProfiler. The edges of organoids were first enhanced using *EnhanceEdges*, followed by identification of organoids as primary objects. The area of the identified organoid objects was then calculated and averaged across every well to obtain the average organoid size per well.

## Statistical analyses

All results were replicated with similar results at frequencies reported in the legends. No statistical methods were used to estimate sample sizes. Researchers were blinded for image analyses as described, otherwise no blinding took place. Statistical analyses of in vivo data were all performed using GraphPad Prism 10. Parametric tests were used, as sample sizes for data were too small to reliably test for normality. For all statistical analyses, a *p*-value < 0.05 was considered significant. All statistical tests and sample sizes for specific panels are stated in the figure legends.

# Data availability

The datasets produced in this study are available in the following databases: -scRNA-seq data: NCBI Gene Expression Omnibus (GEO), accession number GSE267351.

The source data of this paper are collected in the following database record: biostudies:S-SCDT-10_1038-S44319-025-00370-3.

# Peer review information

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

## Acknowledgements

We would like to acknowledge support from the Boston University Single Cell RNA-sequencing Core and Flow Cytometry Facilities. This work was supported in part by funds from the Dahod Grant Program for breast cancer research at Boston University. XV was funded by NIH/NHLBI R01HL124392 and NIH/NIDCR R01DE033519. JK was funded by NIH/NCI grant F31CA232683. AJS was funded by NIH/NIDCR F30AG077929. LK was funded by NIH/NIDCR F31DE033292. SM was in part funded by Find the Cause Breast Cancer Foundation and NIH/NIDCR R01DE031831.

## Author contributions

**Joseph G Kern**: Conceptualization; Data curation; Formal analysis; Validation; Investigation; Visualization; Methodology; Writing—original draft; Writing—review and editing. **Lina Kroehling**: Data curation; Formal analysis; Writing—review and editing. **Anthony J Spinella**: Formal analysis; Validation; Investigation; Visualization. **Stefano Monti**: Resources; Supervision; Methodology; Writing—review and editing. **Xaralabos Varelas**: Conceptualization; Data curation; Supervision; Funding acquisition; Investigation; Writing—original draft; Project administration; Writing—review and editing.

Source data underlying figure panels in this paper may have individual authorship assigned. Where available, figure panel/source data authorship is listed in the following database record: biostudies:S-SCDT-10_1038-S44319-025-00370-3.

## Disclosure and competing interests statement

The authors declare no competing interests.

# Expanded View Figures

**Figure EV1. Epithelial and stromal alterations in basal-like mammary carcinomas driven by LATS1/2 deletion.**

(A) Immunofluorescence staining for DAPI, KRT14, and KRT8 in CTL and LATS1/2-KO mammary glands (Scale bar, 50 μm) and quantification of nuclei counts per area in the mammary stroma ($n = 6$ CTL mice, $n = 6$ L1/2-KO mice, 5 regions analyzed per mouse. Data are shown with mean ± SEM. Two-tailed unpaired T-test). (B) Heatmap of the top 10 differentially expressed genes in each cluster (ranked by Log2FC and using genes expressed in at least 50% of cells in the cluster). Note that if a gene is duplicated in the top 10 genes of two separate clusters, it is only shown once. (C) Violin plot of EYFP expression across all clusters ($n = 3496$ cells for C1, $n = 2666$ cells for C2, $n = 2458$ cells for C3, $n = 1607$ cells for C4, $n = 1483$ cells for C5, $n = 1156$ cells for C6, $n = 1031$ cells for C7, $n = 928$ cells for C8, $n = 799$ cells for C9, $n = 611$ cells for C10, $n = 603$ cells for C11, $n = 440$ cells for C12, $n = 417$ cells for C13, $n = 369$ cells for C14, $n = 140$ cells for C15). (D) UMAP of cell groups specified for CellChat analyses.

▶

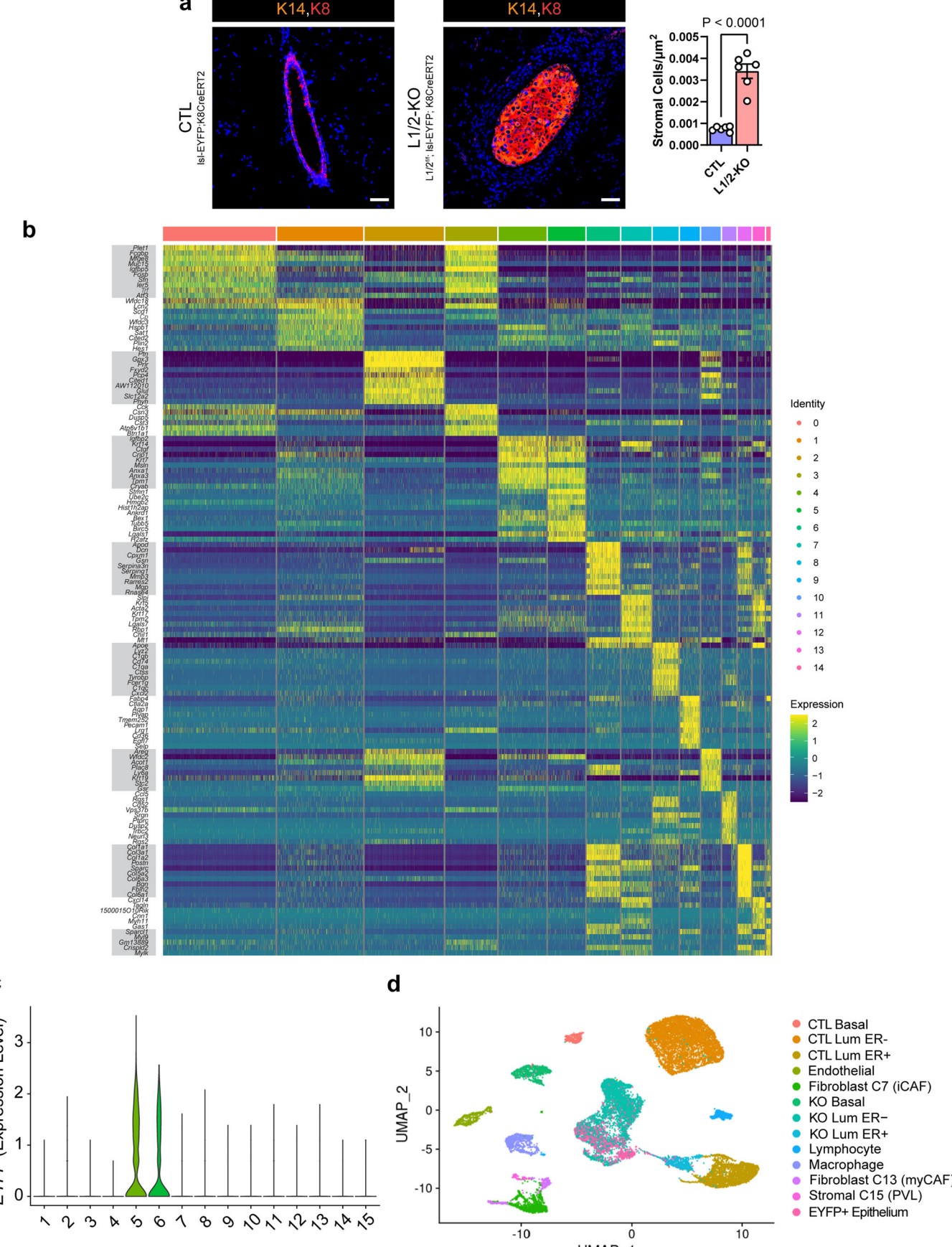

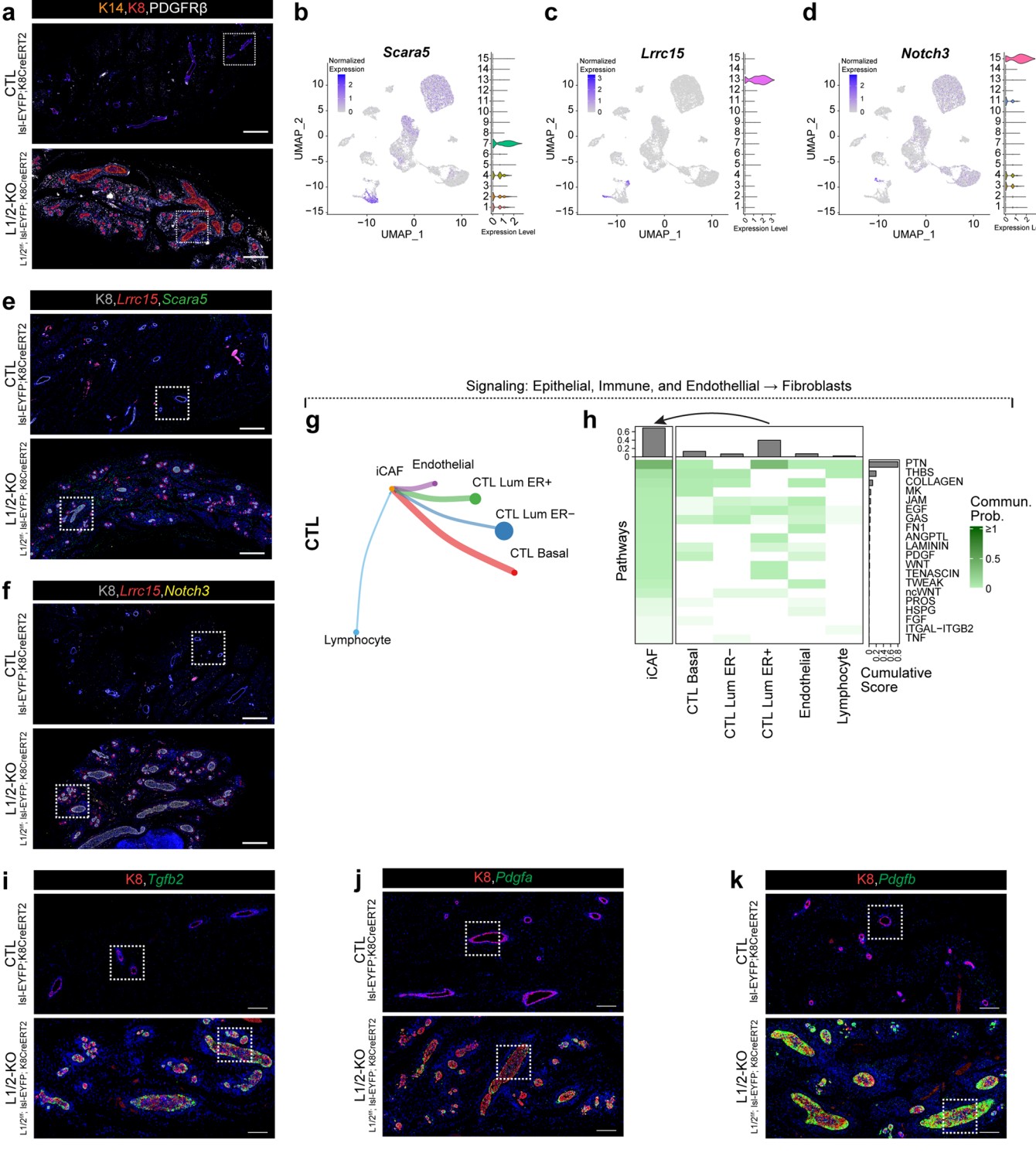

◀ **Figure EV2.** **Accumulation of cancer-associated fibroblasts in the mammary stroma of mice with LATS1/2 inactivation.**

(A) Immunofluorescence staining for KRT14, KRT8, and PDGFRβ in CTL and LATS1/2-KO mammary glands. Dashed boxes indicate subsets shown in Fig. 3A (Scale bar, 500 μm). (B–D) UMAPs of transcriptional markers of the iCAF (*Scara5*) (B), myCAF (*Lrrc15*) (C), and PVL (*Notch3*) (D) populations, along with violin plots of the expression of each gene across all clusters (For violin plots in (B–D), $n = 3496$ cells for C1, $n = 2666$ cells for C2, $n = 2458$ cells for C3, $n = 1607$ cells for C4, $n = 1483$ cells for C5, $n = 1156$ cells for C6, $n = 1031$ cells for C7, $n = 928$ cells for C8, $n = 799$ cells for C9, $n = 611$ cells for C10, $n = 603$ cells for C11, $n = 440$ cells for C12, $n = 417$ cells for C13, $n = 369$ cells for C14, $n = 140$ cells for C15). (E) RNAscope of *Lrrc15* and *Scara5* along with IF for KRT8 in CTL and LATS1/2-KO mammary glands. Dashed boxes indicate subsets shown in Fig. 3E (Scale bar, 500 μm) ($n = 3$ CTL mice, $n = 3$ L1/2-KO mice). (F) RNAscope of *Lrrc15* and *Notch3* along with IF for KRT8 in CTL and LATS1/2-KO mammary glands. Dashed boxes indicate subsets shown in Fig. 3F (Scale bar, 500 μm) ($n = 3$ CTL mice, $n = 3$ L1/2-KO mice). (G) CellChat circle plots showing communication networks from epithelial and stromal cell groups (senders) to iCAF group (receiver) in CTL mammary glands. Line width represents the total number of interactions identified. (H) Heatmap of communication probabilities of signaling pathways from epithelial and non-fibroblast-associated stromal cell groups to the iCAF population in CTL mammary glands. (I–K) RNAscope for *Tgfb2* (I), *Pdgfa* (J), and *Pdgfb* (K) in CTL and LATS1/2-KO mammary glands, along with IF for KRT8. Dashed boxes indicate subsets shown in Fig. 3J–L (Scale bar, 200 μm) ($n = 3$ CTL mice, $n = 3$ L1/2-KO mice each for I-K).

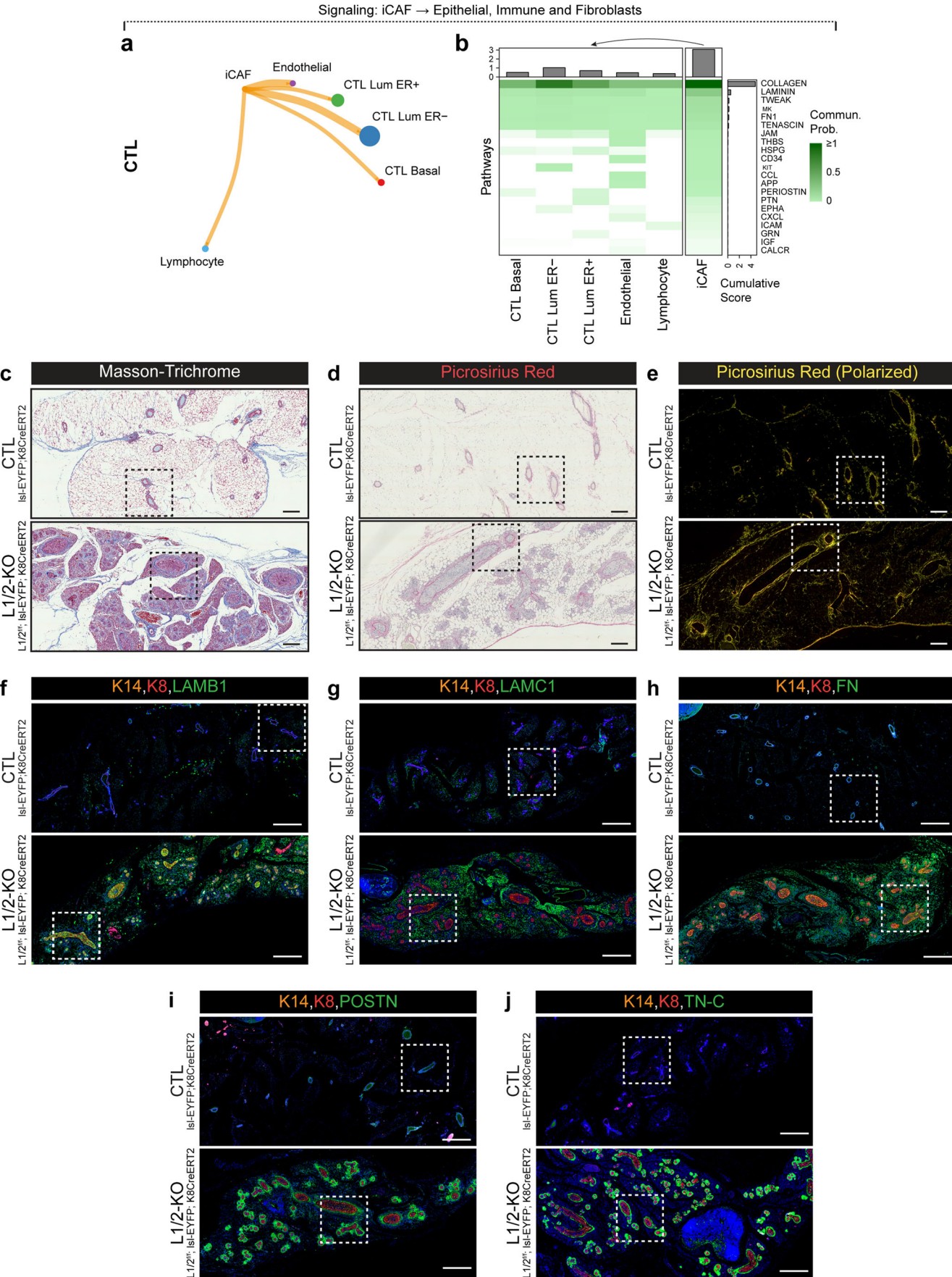

**Figure EV3.  Mammary carcinomas driven by LATS1/2 deletion display deposition of extracellular matrix proteins.**

(A) CellChat circle plots showing communication networks from the iCAF group (senders) to epithelial and stromal cell groups (receivers) in CTL mammary glands. Line widths represent the total number of interactions identified. (B) Heatmap of communication probabilities of signaling pathways from the iCAF group to epithelial and stromal cell groups in CTL mammary glands. (C) Masson-Trichrome staining of CTL and LATS1/2-KO mammary glands (Scale bar 200 µm) ($n = 6$ CTL mice, $n = 4$ L1/2-KO mice from two experiments). (D–E) Picrosirius red staining in CTL and LATS1/2-KO mammary glands using brightfield (D) and polarizing (E) light. (Scale bar 200 µm) ($n = 4$ CTL mice, $n = 6$ L1/2-KO mice). (F–J) Immunofluorescence staining for KRT14, KRT8, and Laminin β1 (F), Laminin γ1 (G), Fibronectin (H), Periostin (I), and Tenascin C (J) in CTL and LATS1/2-KO mammary glands (Scale bar 500 µm) ($n = 6$ CTL mice, $n = 6$ L1/2-KO mice for each panels F–J). For all images, the dashed boxes indicate the subsets that are shown in Fig. 4.

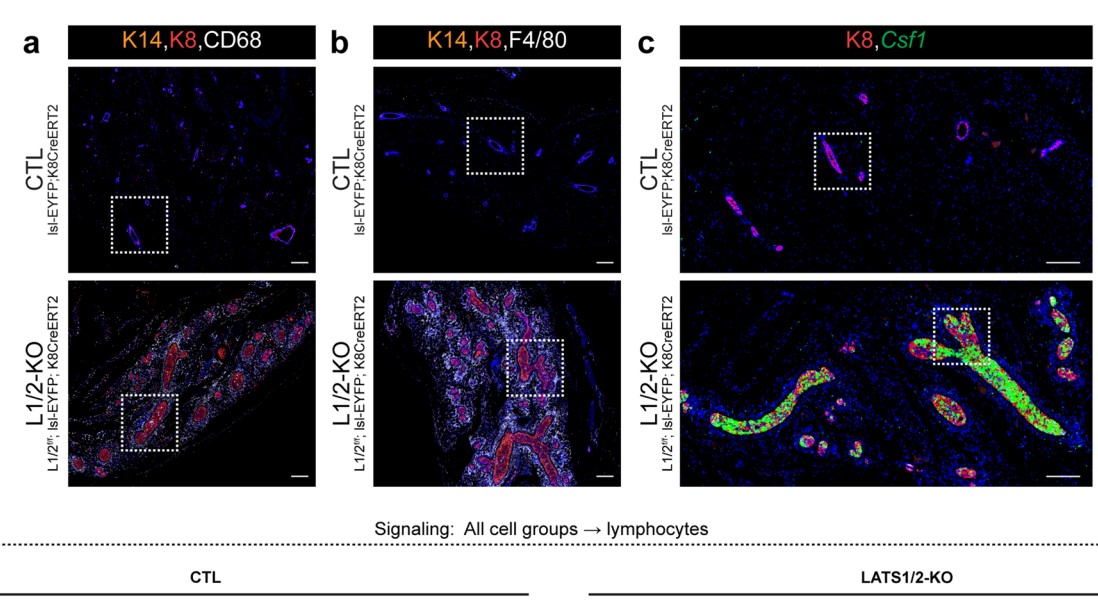

◄ **Figure EV4. Immune cell alterations in carcinomas driven by LATS1/2 inactivation and association of YAP/TAZ-TEAD with epithelial and stromal remodeling in mouse and human breast cancers.**

(A) Immunofluorescence staining for KRT8, KRT14, and CD68 in CTL and LATS1/2-KO mammary glands. Dashed boxes indicate subsets shown in Fig. 5D (Scale bar, 200 μm) (n = 6 CTL mice, n = 6 L1/2-KO mice). (B) Immunofluorescence staining for KRT8, KRT14, and F4/80 in CTL and LATS1/2-KO mammary glands. Dashed boxes indicate subsets shown in Fig. 5E (Scale bar, 200 μm) (n = 6 CTL mice, n = 6 L1/2-KO mice). (C) RNAscope for *Csf1* along with IF for KRT8 in CTL and LATS1/2-KO mammary glands. Dashed boxes indicate subsets shown in Fig. 5I (Scale bar, 200 μm) (n = 3 CTL mice, n = 3 L1/2-KO mice). (D) CellChat circle plots showing communication networks from all cell groups (senders) to lymphocytes (receivers) in CTL and LATS1/2-KO mammary glands. Line widths represent the total number of interactions identified. (E) Heatmap of communication probabilities of signaling pathways from all cell groups to lymphocytes in CTL and LATS1/2-KO mammary glands. (F) GSEA enrichment plots of selected genesets enriched in sorted EYFP$^+$ cells from the mammary glands of LATS1/2-KO (LATS1/2$^{ff}$; lsl-EYFP; Krt8CreERT2) mice relative to CTL (lsl-EYFP; Krt8CreERT2) mice. (G) Dot heatmap of TEAD transcription factors across all single-cell RNA-seq clusters. (H) TIMER Analysis of LATS1/2-KO epithelial signature correlated with presence of stromal cell populations in human TCGA samples. Gray boxes indicate that this cell type was not included in the indicated database. *fdr < 0.05, **fdr < 0.01, ****fdr <0.0001, ns = not significant.

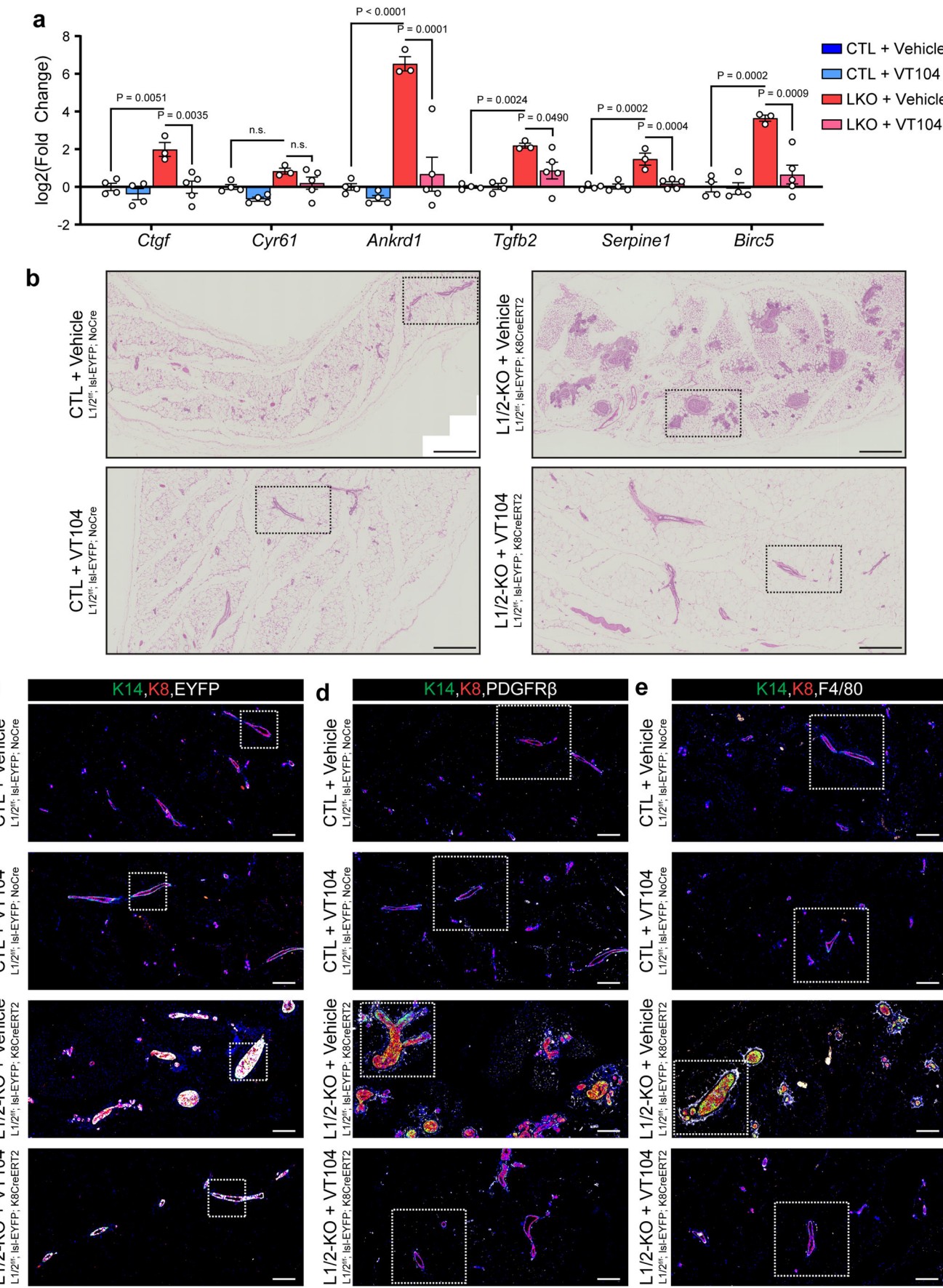

**Figure EV5. Pharmacological inhibition of YAP/TAZ reverses epithelial and stromal alterations in LATS1/2-null carcinomas.**

(A) RT-qPCR analysis of YAP/TAZ-target gene expression in samples of epithelial and stromal cells collected from the 3rd and 5th mammary glands of CTL (Lats1/2$^{f/f}$; lsl-EYFP; NoCre) and L1/2-KO (Lats1/2$^{f/f}$; lsl-EYFP; K8CreERT2) mice treated with Vehicle or VT104 as outlined in Fig. 7A ($n = 4$ CTL + Vehicle mice, $n = 4$ CTL + VT104 mice, $n = 3$ L1/2-KO + Vehicle mice, $n = 5$ L1/2-KO + VT104 mice. Data are shown with mean ± SEM. Ordinary two-way ANOVAs with Tukey's multiple comparisons tests. Adjusted $P$-values are displayed in figure). (B) Hematoxylin and eosin stain of CTL (LATS1/2$^{ff}$; lsl-EYFP; noCre) and LATS1/2-KO (LATS1/2$^{ff}$; lsl-EYFP; Krt8CreERT2) mammary glands treated with Vehicle and VT104. Dashed boxes indicate subsets shown in Fig. 7A (Scale bar, 500 μm). (C) Immunofluorescence staining for KRT8, KRT14, and EYFP in mammary glands of CTL and LATS1/2-KO mice treated with Vehicle and VT104. Dashed boxes indicate subsets shown in Fig. 7B (Scale bar, 200 μm). (D) Immunofluorescence staining for KRT14, KRT8, and PDGFRβ in mammary glands of CTL and LATS1/2-KO mice treated with VT104. Dashed boxes indicate subsets shown in Fig. 7C (Scale bar, 200 μm). (E) Immunofluorescence staining for KRT14, KRT8, and F4/80 in mammary glands of CTL and LATS1/2-KO mice treated with VT104. Dashed boxes indicate subsets shown in Fig. 7D (Scale bar, 200 μm). For all panels: $n = 4$ CTL + Vehicle mice, $n = 4$ CTL + VT104 mice, $n = 3$ L1/2-KO + Vehicle mice, $n = 5$ L1/2-KO + VT104 mice.

