## [Peer Review File · EMBO Reports]

LATS1/2 Inactivation in the Mammary Epithelium Drives the Evolution of a Tumor-Associated Niche

Joseph Kern, Lina Kroehling, Anthony Spinella, Stefano Monti, and Xaralabos Varelas

Corresponding author(s): Xaralabos Varelas (xvarelas@bu.edu)

Review Timeline:

Submission Date:	10th Jun 24
Editorial Decision:	8th Jul 24
Revision Received:	7th Nov 24
Editorial Decision:	18th Nov 24
Revision Received:	2nd Jan 25
Accepted:	8th Jan 25

Editor: Achim Breiling

Transaction Report:

Dear Prof. Varelas,

Thank you for the submission of your manuscript to EMBO reports. I have now received the reports from the three referees that were asked to evaluate your study, which can be found at the end of this email. As you will see, the referees have several comments, concerns, and suggestions, indicating that a major revision of the manuscript is necessary to allow publication of the study in EMBO reports. As the reports are below, and all the concerns need to be addressed, I will not detail them further here.

Given the constructive referee comments, I would like to invite you to revise your manuscript with the understanding that the concerns of the referees must be addressed in the revised manuscript or in a detailed point-by-point response. Acceptance of your manuscript will depend on a positive outcome of a second round of review. It is EMBO reports policy to allow a single round of revision only and acceptance of the manuscript will therefore depend on the completeness of your responses included in the next, final version of the manuscript.

1) a .docx formatted version of the final manuscript text (including legends for main figures, EV figures and tables), but without the figures included. Figure legends should be compiled at the end of the manuscript text.

2) individual production quality figure files as .eps, .tif, .jpg (one file per figure), of main figures and EV figures. Please upload these as separate, individual files upon re-submission.

4) a complete author checklist, which you can download from our author guidelines

(<https://www.embopress.org/page/journal/14693178/authorguide>). Please insert page numbers in the checklist to indicate where the requested information can be found in the manuscript. The completed author checklist will also be part of the RPF.

5) that primary datasets produced in this study (e.g. RNA-seq, ChIP-seq, structural and array data) are deposited in an appropriate public database. If no primary datasets have been deposited, please also state this in a dedicated section (e.g. 'No primary datasets have been generated and deposited'), see below.

The accession numbers and database should be listed in a formal "Data Availability" section (placed after Materials & Methods) that follows the model below. This is now mandatory (like the COI statement). Please note that the Data Availability Section is restricted to new primary data that are part of this study. This section is mandatory. As indicated above, if no primary datasets have been deposited, please state this in this section

Data availability

8) Regarding data quantification and statistics, please make sure that the number "n" for how many independent experiments were performed, their nature (biological versus technical replicates), the bars and error bars (e.g. SEM, SD) and the test used to calculate p-values is indicated in the respective figure legends (also for EV figures and all those in an Appendix). Please also check that all the p-values are explained in the legend, and that these fit to those shown in the figure. Please provide statistical testing where applicable. Please avoid the phrase 'independent experiment', but clearly state if these were biological or technical replicates. Please also indicate (e.g. with n.s.) if testing was performed, but the differences are not significant. In case n=2, please show the data as separate datapoints without error bars and statistics. See also: <http://www.embopress.org/page/journal/14693178/authorguide#statisticalanalysis>

9) Please add scale bars of similar style and thickness to microscopic images, using clearly visible black or white bars (depending on the background). Please place these in the lower right corner of the images themselves. Please do not write on or near the bars in the image but define the size in the respective figure legend.

10) Please also note our reference format:

12) We now use CRediT to specify the contributions of each author in the journal submission system. CRediT replaces the author contribution section. Please use the free text box to provide more detailed descriptions and do NOT provide your final manuscript text file with an author contributions section. See also our guide to authors: <https://www.embopress.org/page/journal/14693178/authorguide#authorshipguidelines>

13) All Materials and Methods need to be described in the main text using our 'Structured Methods' format, which is required for all research articles. According to this format, the Materials and Methods section should include a Reagents and Tools Table (listing key reagents, experimental models, software, and relevant equipment and including their sources and relevant

identifiers), uploaded as separate file, followed by a Methods and Protocols section in which we encourage the authors to describe their methods using a step-by-step protocol format with bullet points, to facilitate the adoption of the methodologies across labs. More information on how to adhere to this format as well as downloadable templates (.doc) for the Reagents and Tools Table can be found in our author guidelines (section 'Structured Methods'):

14) Please add 5 keywords to the manuscript text file and order the manuscript sections like this, using these names: Title page - Abstract - Keywords - Introduction - Results - Discussion - Methods - Data availability section - Acknowledgements - Disclosure and Competing Interests Statement - References - Figure legends - Expanded View Figure legends

I look forward to seeing a revised version of your manuscript when it is ready. Please let me know if you have questions or comments regarding the revision.

Please use this link to submit your revision: <https://embor.msubmit.net/cgi-bin/main.plex>

Yours sincerely,

Referee #1:

The proposed manuscript "LATS1/2 Inactivation in the Mammary Epithelium Drives the Evolution of a Tumor- Associated Niche" by Kern et al. aims to demonstrate the profound remodeling at single-cell level of both the epithelial and stromal compartment after specific depletion of LATS1/2. In their previous work (Kern et al., 2022), the authors showed that Lats1/2-null cells exhibit luminal-basal plasticity. Here, they focus on evaluating stromal changes and interaction between stroma and epithelium signaling using CellChat and multiplexed RNAscope in situ hybridization approaches. The authors observed an increase in cancer-associated fibroblast and macrophages populations together with a different localization. Finally, pharmacological inhibition of TEAD activity reverses epithelial and stromal phenotypes. In my opinion, the proposed idea is interesting and globally supported by experimental evidence. Data are well-described and organized. I would suggest to the authors some additional insights to improve the quality of the manuscript and broad its impact.

Specific comments

- In the introduction authors affirmed that basal- like breast cancer is "largely" synonymous of triple-negative breast cancer. Even though many tumours expressing basal markers are triple- negative, however, the contrary is not always true, as shown in numerous studies. The authors need to better express this concept.

-LATS1/2 deletion specifically in epithelial breast cancer cells of mice models nicely shows cells and surrounding stroma remodeling. To strengthen the clinical readout of these data, could the authors show this in patients' samples? One possibility could be represented by existing breast cancer datasets in which LATS1/2 expression can be evaluated and associated with stromal signatures.

Referee #2:

The authors investigate the Hippo-YAP/TAZ pathway in tumor-stromal microenvironment in the context of basal like breast cancer. They find that LATS1/2 CKO basal-like carcinoma initiation has distinct cancer-associated fibroblast and macrophage influx with extracellular matrix remodeling, a phenotype that has similarities to human triple-negative breast tumors. Overall, an interesting and well written paper with solid data that benefit from a bit more functional supportive data.

Comments:

1. "Dysregulated YAP/TAZ has been associated with increased tumor elasticity (Dupont et al, 2011)"
"Elasticity" seems to be awkward word usage
2. Data in figure 1A should be quantified by measuring and performing stats on the colocalization of relevant cell types in mutant vs control
3. I wonder if there is a better way to display figure 2A. Dot plots certainly have value, but the authors may consider trying box plots or heat maps to more effectively convey the message.
4. The cellchat data can be improved by simplifying and focusing on the most relevant predicted cell-cell interactions. This is important as it is central to the paper.
5. The connection to the human data is very strong as presented in figure 3.
6. Can the authors provide more supportive evidence about the outgoing signaling factors from epithelium which are purported to be encoded by YAP/TAZ target genes. Some type of regulon analysis might be helpful (ie. SCENIC).
7. The use of the Tead palmitoylation inhibitor is a good addition in terms of functional data. However, the authors should also look at inhibiting some of the proposed downstream signaling events to provide more functional support for the model. Currently the paper relies very heavily on profiling and predictive approaches.

Referee #3:

Kern et al investigate the consequences of inactivating LATS1/2 in mammary epithelium. This study follows a 2022 Nat Comm manuscript from the same group that showed that loss of Lats1/2 drives luminal - basal plasticity and initiation of basal-like breast tumors. In the present study the authors exploit a Keratin8 driven Cre to delete Lats1/2 in luminal epithelial cells (similar to the 2022 paper) and then follow changes in the mammary gland microenvironment through histology and scRNAseq. This is somewhat similar to the 2022 Nat Comm paper; however in the present study bioinformatics, immunofluorescence and inhibition of TEAD transcription factor in vivo are employed to support the conclusions drawn. The current work is distinguished from the prior study through its focus on stromal changes and identification of potential cell-cell communication pathways in the process of tumor initiation. In general the article is presented well. The text is clear and the figures are cogent and presented in a logical manner. The work is analysis of what appears to be two animal experiments from which the authors conclude that Hippo pathway inactivation in luminal epithelial cells drives the development of a basal-like tumor microenvironment dependent on dysregulated cell-cell communication. Further the authors suggest that inhibition of the dysregulated YAP/TAZ activity may have therapeutic potential in basal-like breast cancer.

Comments:

1. Based on network and CellChat analysis, the manuscript provides many intriguing candidates that might drive functionally important cell-cell communication networks the mammary microenvironment but does not provide validation of any. Protein expression of some targets is shown but there is no functional demonstration that any of the pathways are critical to the observed phenotype, this is a weakness that detracts from the impact of the study. Each and every potential pathway highlighted in the CellChat data need not be functionally interrogated and none need to be validated in vivo (although that would be ideal). In vitro studies showing that factor X (PTN, THBS, collagen etc) has a relevant effect on a relevant target cell in vitro and can be blocked with an inhibitor or KD would suffice.
2. The paper demonstrates that blocking the TEAD transcription factors, major effectors of YAP/TAZ, can rescue the phenotype of KO of Lats1/2 in Krt8+ cells. This is not surprising. The authors mention that TEAD inhibition inhibits the growth of Nf2-mutated mesothelioma cells and are currently being tested clinically. A) Has VT104 or other related strategies been used in other models of basal-like breast tumors and if so was therapy effective, did it depend on treating tumors at an early stage? B) Does treatment of VT104 alter the expression of candidate pathways identified in the CellChat analysis?

RESPONSE TO REVIEWERS' COMMENTS

We thank all the reviewers for their positive assessment of the manuscript and constructive comments. We have listed a point-by-point response below to each of the comments (listed in blue).

Referee #1:

The proposed manuscript "LATS1/2 Inactivation in the Mammary Epithelium Drives the Evolution of a Tumor- Associated Niche" by Kern et al. aims to demonstrate the profound remodeling at single-cell level of both the epithelial and stromal compartment after specific depletion of LATS1/2. In their previous work (Kern et al., 2022), the authors showed that Lats1/2-null cells exhibit luminal-basal plasticity. Here, they focus on evaluating stromal changes and interaction between stroma and epithelium signaling using CellChat and multiplexed RNAscope in situ hybridization approaches. The authors observed an increase in cancer-associated fibroblast and macrophages populations together with a different localization. Finally, pharmacological inhibition of TEAD activity reverses epithelial and stromal phenotypes. In my opinion, the proposed idea is interesting and globally supported by experimental evidence. Data are well-described and organized. I would suggest to the authors some additional insights to improve the quality of the manuscript and broad its impact.

Response: We thank the reviewer for their positive opinion. As described below, we have clarified our text to better describe the biology of triple-negative breast cancer and have added additional data linking our observations to stromal phenotypes in human breast tumors.

Specific comments

- In the introduction authors affirmed that basal- like breast cancer is "largely" synonymous of triple-negative breast cancer. Even though many tumours expressing basal markers are triple-negative, however, the contrary is not always true, as shown in numerous studies. The authors need to better express this concept.

Response: We have reworked this section of our introduction to eliminate this confusion and better emphasize that most, but not all, triple-negative tumors are basal-like, adding references to support this.

-LATS1/2 deletion specifically in epithelial breast cancer cells of mice models nicely shows cells and surrounding stroma remodeling. To strengthen the clinical readout of these data, could the authors show this in patients' samples? One possibility could be represented by existing breast cancer datasets in which LATS1/2 expression can be evaluated and associated with stromal signatures.

Response: We thank the reviewer for bringing up this point and we have added new analyses of human breast cancer data that support our conclusions that increased YAP/TAZ-TEAD transcriptional activity associates with stromal changes. For this, we performed a TIMER analysis (computational platform that estimates stromal cell infiltration based on TCGA RNA-sequencing data) on the gene signature obtained from epithelial cells of LATS1/2-KO mice. This

revealed strong correlation of the LATS1/2-KO epithelial signature with the presence of cancer-associated fibroblasts and macrophages in human breast cancers, similar to that observed in our mouse model (new Fig EV4h).

Referee #2:

The authors investigate the Hippo-YAP/TAZ pathway in tumor-stromal microenvironment in the context of basal like breast cancer. They find that LATS1/2 CKO basal-like carcinoma initiation has distinct cancer-associated fibroblast and macrophage influx with extracellular matrix remodeling, a phenotype that has similarities to human triple-negative breast tumors. Overall, an interesting and well written paper with solid data that benefit from a bit more functional supportive data.

Response: We thank the reviewer for the positive assessment of our manuscript. As listed below, we have updated aspects of our manuscript based on the suggestions and have added new experimental data that functionally extend our conclusions and offer more insight into YAP/TAZ-TEAD-mediated signals communicating between the epithelium and the stromal niche.

Comments:

1. "Dysregulated YAP/TAZ has been associated with increased tumor elasticity (Dupont et al, 2011)" "Elasticity" seems to be awkward word usage

Response: We have modified the term "Elasticity" to "Stiffness" and have reworked this part of the introduction to make our point clearer.

2. Data in figure 1A should be quantified by measuring and performing stats on the colocalization of relevant cell types in mutant vs control

Response: We have quantified the stromal nucleation surrounding carcinoma lesions that develop in LATS1/2-KO mammary glands compared to controls. For this, we quantified non-epithelial cell (Krt14/Krt8-negative cell) numbers that are located adjacent to the mammary ducts (new Figure EV1b), which confirms stromal cell influx following LATS1/2 inactivation.

3. I wonder if there is a better way to display figure 2A. Dot plots certainly have value, but the authors may consider trying box plots or heat maps to more effectively convey the message.

Response: We agree that a heatmap is a more informative way of showing the data in this panel. We have now updated Fig 2A as suggested.

4. The cellchat data can be improved by simplifying and focusing on the most relevant predicted cell-cell interactions. This is important as it is central to the paper.

Response: We appreciate the comment from the reviewer, as it is often difficult to extract the most relevant data from cell-cell interactions identified in single-cell RNA-sequencing. We have tried making a few alternate iterations of this figure but believe that the current version is the most informative method to communicate the data. To help the reader extract more information from our data, we have now also added a comprehensive table with statistics for all the CellChat signaling pathways identified from our analyses, which is included as Dataset EV1.

5. The connection to the human data is very strong as presented in figure 3.

Response: We thank the reviewer for this comment and we agree that our findings connect well to human data.

6. Can the authors provide more supportive evidence about the outgoing signaling factors from epithelium which are purported to be encoded by YAP/TAZ target genes. Some type of regulon analysis might be helpful (ie. SCENIC).

Response: As suggested, we have performed SCENIC analysis on epithelial cells from our LATS1/2-KO single-cell RNA-sequencing data. This analysis identified the TEAD1 regulon among the top differentially expressed regulons in epithelial cells of LATS1/2-KO mammary glands relative to controls (new Fig 6h), which supports our conclusions and the rationale for therapeutically targeting TEADs. This analysis furthermore identified major outgoing signaling factors we identified in our study, including *Tgfb2*, *Pdgfa*, *Pdgfb*, and *Csfl*, as members of the TEAD1 regulon in these cells.

7. The use of the Tead palmitoylation inhibitor is a good addition in terms of functional data. However, the authors should also look at inhibiting some of the proposed downstream signaling events to provide more functional support for the model. Currently the paper relies very heavily on profiling and predictive approaches.

Response: We appreciate the reviewer's positive assessment of our use of the TEAD palmitoylation inhibitor in this study. Our rationale for targeting TEADs was based on our observations that suggested YAP/TAZ activation throughout the local carcinoma niche that develops within our model. Being that YAP/TAZ-TEAD are likely the key mediators of this niche, we hypothesized that targeting these factors directly would be the most beneficial direction for therapy. We also appreciate the reviewer's further point that identifying additional signals contributing to the dysregulated mammary tumor niche would benefit the conclusions of the manuscript. To further explore this, we focused on focal adhesion signaling, the top pathway enriched in epithelial cells from LATS1/2-KO mice relative to controls. Focal adhesion signaling is regulated by extracellular matrix cues such as those we predicted to signal from stromal cells to the epithelium. We have added new data (Fig 6E) showing that inhibition of Focal Adhesion Kinase (FAK) can reduce the growth of LATS1/2-KO mammary organoids *ex vivo*, suggesting that increased FAK signaling in LATS1/2-KO epithelial cells contributes to pro-tumorigenic growth.

Referee #3:

Kern et al investigate the consequences of inactivating LATS1/2 in mammary epithelium. This study follows a 2022 Nat Comm manuscript from the same group that showed that loss of Lats1/2 drives luminal - basal plasticity and initiation of basal-like breast tumors. In the present study the authors exploit a Keratin8 driven Cre to delete Lats1/2 in luminal epithelial cells (similar to the 2022 paper) and then follow changes in the mammary gland microenvironment through histology and scRNAseq. This is somewhat similar to the 2022 Nat Comm paper; however in the present study bioinformatics, immunofluorescence and inhibition of TEAD transcription factor in vivo are employed to support the conclusions drawn. The current work is distinguished from the prior study through its focus on stromal changes and identification of potential cell-cell communication pathways in the process of tumor initiation. In general the article is presented well. The text is clear and the figures are cogent and presented in a logical manner. The work is analysis of what appears to be two animal experiments from which the authors conclude that Hippo pathway inactivation in luminal epithelial cells drives the development of a basal-like tumor microenvironment dependent on dysregulated cell-cell communication. Further the authors suggest that inhibition of the dysregulated YAP/TAZ activity may have therapeutic potential in basal-like breast cancer.

Response: We thank the reviewer for the comments, as these have helped make our manuscript and conclusions stronger. As outlined below, we modified our manuscript to clarify points that were raised, and we have added additional functional data that extend and strengthen our conclusions.

Comments:

Based on network and CellChat analysis, the manuscript provides many intriguing candidates that might drive functionally important cell-cell communication networks the mammary microenvironment but does not provide validation of any. Protein expression of some targets is shown but there is no functional demonstration that any of the pathways are critical to the observed phenotype, this is a weakness that detracts from the impact of the study. Each and every potential pathway highlighted in the CellChat data need not be functionally interrogated and none need to be validated in vivo (although that would be ideal). In vitro studies showing that factor X (PTN, THBS, collagen etc) has a relevant effect on a relevant target cell in vitro and can be blocked with an inhibitor or KD would suffice.

Response: We have performed additional experiments to better understand signals from the stromal niche that contribute to carcinoma development upon LATS1/2 inactivation. Specifically, we focused on testing the top signaling pathway enriched in LATS1/2-KO epithelia, which is Focal Adhesion Kinase (FAK) signaling. We found that PF-573228, a small molecule inhibitor of FAK, reduced the growth of LATS1/2-KO organoids (new Fig 6e), suggesting that signals regulating FAK that are induced within the LATS1/2-KO carcinomas act as potent growth mediators.

2. The paper demonstrates that blocking the TEAD transcription factors, major effectors of YAP/TAZ, can rescue the phenotype of KO of Lats1/2 in Krt8+ cells. This is not surprising. The

authors mention that TEAD inhibition inhibits the growth of Nf2-mutated mesothelioma cells and are currently being tested clinically. A) Has VT104 or other related strategies been used in other models of basal-like breast tumors and if so was therapy effective, did it depend on treating tumors at an early stage? B) Does treatment of VT104 alter the expression of candidate pathways identified in the CellChat analysis?

Response: To our knowledge this is the first use of a TEAD palmitoylation inhibitor in a basal-like breast cancer model, and the first use in a LATS1/2-KO genetic model of mammary tumors. As suggested by the reviewer we have extended our analysis of the consequences of TEAD inhibition on the LATS1/2-KO environment by examining signaling effectors induced in the epithelium following LATS1/2-inactivation. We have added new data (Fig 7E-G) demonstrating a reduction of *Tgfb2*, *Pdgfb*, and *Csfl* in the epithelium of LATS1/2-KO mammary glands following TEAD inhibition, which are effectors that have known roles for modulating fibroblast and macrophage responses in tumors. These data support our conclusions that TEAD inhibitors block stromal remodeling in developing carcinomas resulting from LATS1/2 inactivation and provide additional rationale for the use of this class of inhibitors in targeting breast tumors with high YAP/TAZ-TEAD activity.

Dear Prof. Varelas

Thank you for the submission of your revised manuscript to our editorial offices. I have now received the reports from the referees that I asked to re-evaluate the study, you will find below. As you will see, all three referees now support publication of the study in EMBO reports.

Before I can proceed with formal acceptance, I have these editorial requests I ask you to address in a final revised manuscript:

- Please add up to 5 keywords to the manuscript and put these below the abstract.
- We now use CRediT to specify the contributions of each author in the journal submission system. CRediT replaces the author contribution section. Please use the free text box to provide more detailed descriptions and do NOT provide your final manuscript text file with an author contributions section. See also our guide to authors: <https://www.embopress.org/page/journal/14693178/authorguide#authorshipguidelines>
- Please add scale bars of similar style and thickness to all microscopic images, using clearly visible black or white bars (depending on the background). Please place these in the lower right corner of the images themselves. Please do not write on or near the bars in the image but define the size in the respective figure legend. Presently, several scale bars are too thin or placed elsewhere. Please check.
- Please check again that the number "n" for how many independent experiments were performed, their nature (biological versus technical replicates), the bars and error bars (e.g. SEM, SD) and the test used to calculate p-values is indicated in the respective figure legends (main, EV and Appendix figures). Please also check that all the p-values are explained in the legend, and that these fit to those shown in the figure. Please provide statistical testing where applicable. Please avoid the phrase 'independent experiment', but clearly state if these were biological or technical replicates. Please also indicate (e.g. with n.s.) if testing was performed, but the differences are not significant. In case n=2, please show the data as separate datapoints without error bars and statistics. See also: <http://www.embopress.org/page/journal/14693178/authorguide#statisticalanalysis>

- It seems that the information related to n is missing in the legends of figures 3d; 6f; EV 1c; EV 2b-d; EV 5a. Please check.
- The data citation "Data Ref Kern et al., 2022" does not refer to deposited experimental data, but refers to journal article. Please check
- Please add the primer information provided in the Methods section to the Reagents and tools table and add a callout (e.g. 'see reagents and tools table').
- Please remove the referee token from The Data Availability section (DAS) and add a direct link to the deposited dataset. Moreover, please make sure the dataset GSE267351 is public latest for the date of online publication of the study.
- I noted that all subsets of microscopic images marked with dashed boxes in the EV figures are also shown in the main figures. This is also indicated for most cases in the EV figure legends. However, it seems there is no mention that the image shown in 5A lower left (L1/2-KO) is the same as EV4B bottom (L1/2-KO). Moreover, the indication is missing that the dashed boxes in EV4A are also shown in 5D and that the dashed boxes in EV4B are shown in 5E. Please carefully check that each panel reuse is appropriately indicated in the final figure legends.

In addition, I would need from you uploaded separately:

Please use this link to submit your revision: <https://embor.msubmit.net/cgi-bin/main.plex>

Besy,

Referee #1:

The authors have adequately addressed to the concerns raised previously by this reviewer. Based on this, the manuscript in its present form is suitable for publication in EMBO Reports.

Referee #2:

The concerns have been addressed - this is overall a nice piece of work.

Referee #3:

The authors have adequately addressed the comments of the initial review. It is an interesting and well done study.

All editorial and formatting issues were resolved by the authors.

Prof. Xaralabos Varelas
Boston University School of Medicine
Biochemistry
72 East Concord Street
Room K620
Boston, MA 02118
United States

Dear Prof. Varelas,

I am very pleased to accept your manuscript for publication in the next available issue of EMBO reports. Thank you for your contribution to our journal.

Yours sincerely,
